# Time-lapse monitoring of root water uptake using electrical resistivity tomography and Mise-à-la-Masse: a vineyard infiltration experiment

Benjamin Mary[1], Luca Peruzzo[2,3], Jacopo Boaga[1], Nicola Cenni[1], Myriam Schmutz[3], Yuxin Wu[2], Susan S. Hubbard[2], Giorgio Cassiani[1]

[1]Dipartimento di Geoscienze, Università degli Studi di Padova, Via G. Gradenigo, 6–35131 Padova, Italy
[2]Earth and Environmental Sciences Area, Lawrence Berkeley National Laboratory, 1 Cyclotron Rd, Berkeley, CA, 94720, USA.
[3]EA G&E 4592, Bordeaux INP, University Bordeaux Montaigne, 1 allée Daguin, 33607 Pessac, France

Correspondence to: Benjamin Mary (benjamin.mary@unipd.it)

**Abstract.** This paper presents a time-lapse application of electrical methods (Electrical Resistivity Tomography – ERT – and Mise-à-la-Masse – MALM) for monitoring plant roots and their activity (root water uptake) during a controlled infiltration experiment. The use of non-invasive geophysical monitoring is of increasing interest as these techniques provide time-lapse imaging of processes that otherwise can only be measured at few specific spatial locations. The experiment here described was conducted in a vineyard in Bordeaux (France) and was focused on the behaviour of two neighbouring grapevines. The joint application of ERT and MALM has several advantages. While ERT in time-lapse mode is sensitive to changes in soil electrical resistivity and thus to the factors controlling it (mainly soil water content, in this context), MALM uses DC current injected in a tree stem to image where the plant-root system is in effective electrical contact with the soil at locations that are likely to be the same where root water uptake (RWU) takes place. Thus, ERT and MALM provide complementary information about the root structure and activity. The experiment shows that the region of likely electrical current sources produced by MALM does not change significantly during the infiltration time in spite of the strong changes of electrical resistivity caused by changes in soil water content. Ultimately, the interpretation of the current source distribution strengthened the hypothesis of using current as a proxy for root detection. This fact, together with the evidence that current injection in the soil and in the stem produce totally different voltage patterns, corroborates the idea that this application of MALM highlights the active root density in the soil. When considering the electrical resistivity changes (as measured by ERT) inside the stationary volume of active roots delineated by MALM, the overall tendency is towards a resistivity increase during irrigation time, which can be linked to a decrease in soil water content caused by root water uptake. On the contrary, when considering the soil volume outside the MALM-derived root water uptake region, the electrical resistivity tends to decrease as an effect of soil water content increase caused by the infiltration. The use of a simplified infiltration model confirms at least qualitatively this behaviour. The monitoring results are particularly promising, and the method can be applied to a variety of scales including the laboratory scale where direct evidence of roots structure and root water uptake can help corroborate the approach. Once fully validated, the joint use of MALM and ERT can be used as a valuable tool to study the activity of roots under a wide variety of field conditions.

## 1 Introduction

The interaction between soil and biota is one of the main mechanisms controlling the exchange of mass and energy between the Earth's terrestrial ecosystems and the atmosphere. Philip (1966) was the first to use the phrase "soil–plant–atmosphere continuum" (SPAC) to conceptualize this interface in the framework of continuum physics. Even though more than five decades have elapsed and many efforts have been expanded (e.g., Maxwell et al., 2007; de Arellano et al., 2012; Anderegg et al., 2013; Band et al., 2014), the current mechanistic understanding or modelling of SPAC is still unsatisfactory (e.g. Dirmeyer et al., 2006, 2014 and Newman et al., 2006). This is not totally surprising, since soil-plant interactions are complex, exhibiting scale- and species-dependence with high soil heterogeneity and plant growth plasticity. In this study, we focus on new methods

designed to image root systems and their macroscopic functioning, in order to help understand the complex mechanisms of
these systems (the rhizosphere, e.g. York et al., 2016). This diversity of interactions presents an enormous scientific challenge
to understanding the linkages and chain of impacts (Richter and Mobley, 2009).
Roots contribute substantially to carbon sequestration. Roots are the connection between the soil, where water and nutrients
reside, to the other organs and tissues of the plant, where these resources are used. Hence roots provide a link in the pathway
for fluxes of soil water and other substances through the plant canopy to the atmosphere (e.g. Dawson and Stiegwolf, 2007).
These transpiration fluxes are responsible for the largest fraction of water leaving the soil in vegetated systems (Chahine,
1992). Root Water uptake (RWU) influences the water dynamics in the rhizosphere (Couvreur et al., 2012) and the partitioning
of net radiation into latent and sensible heat fluxes thereby impacting atmospheric boundary layer dynamics (Maxwell et al.,
2007; de Arellano et al., 2012). Yet, a number of issues remain when representing RWU in both hydrological and atmospheric
models. Dupuy et al. (2010) summarize the development of root growth models from its origins in the 1970s with simple
spatial models (Hackett and Rose, 1972; Gerwitz and Page, 1974) to the development of very complex plant architectural
models (Jourdan and Rey, 1997). Dupuy et al. (2010) advocate for a different approach, where roots systems are described as
"density" distributions. Attempts in this direction (Dupuy et al., 2005; Draye et al., 2010; Dupuy and Vignes, 2012) require
much less specific knowledge of the detailed mechanisms of meristem evolution, and yet are sufficient to describe the root
"functions" in the framework of continuum physics, i.e. the one endorsed by the SPAC concept. These models also lend
themselves more naturally to calibration against field evidence, as they focus on the "functioning" of roots, especially in terms
of RWU (e.g. Volpe et al., 2013, Manoli et al., 2014). However, calibration requires that suitable data such as root density and
soil water content evolution are available in a form comparable with the model to be calibrated. This is the main motivation
behind the work presented herein.
A thorough understanding of root configuration in space and their evolution in time is impossible to achieve using only
traditional invasive methods: this is particularly true for root hairs, i.e. for the absorptive unicellular extensions of epidermal
cells of a root. These tiny, hair-like structures function as the major site of water and mineral uptake. Root hairs are extremely
delicate, turn over quickly, and are subject to desiccation and easily destroyed. For these reasons, direct investigation of their
in situ structure via excavation is practically impossible under field conditions.
The development of non-invasive or minimally invasive techniques is required to overcome the limitations of conventional
invasive characterization approaches. Non-invasive methods are based on physical measurements at the boundary of the
domain of interest, i.e. at the ground surface and, when possible, in shallow boreholes. Non-invasive methods provide spatially
extensive, high-resolution information that can also be supported by more traditional local and more invasive data such as soil
samples, TDR, lysimeters and rhizotron measurements.
Electrical signals may contribute to the detection of roots and to the characterization of their activities. For instance, self-
potential (SP) signals can be associated with plant activities: water uptake generates a water circulation and a mineral
segregation at the soil–roots interface that induce ionic concentration gradients which in turn generate voltages of the order of
a few mV (Gibert et al., 2006). However, such SP sources are generally too low to be detectable in normally noisy environment.
Induced Polarization (e.g. Kemna et al., 2012) is also a promising approach in root monitoring. This is consistent with the fact
that root systems are commonly modelled as electrical circuits composed of resistance R and capacitance C (e.g. Dalton, 1995
and similar models). Recently, Mary et al. (2017) considered polarization from soil to root tissues, as well as the polarization
processes along and around roots, to explain the phase shift (between injected current and voltage response) observed for
different soil water content. Weigand and Kemna (2017, 2019) demonstrated that multi-frequency electrical impedance
tomography is capable of imaging root systems extent.
In the investigation of roots and RWU the most widely used non-invasive technique is Electrical Resistivity Tomography (ERT
– e.g. Binley and Kemna, 2005). ERT measures soil electrical resistivity and, in time-lapse mode, resistivity changes over
time. Electrical resistivity values depend on soil type and its porosity, but also on state variables such as the saturation of

electrolyte (water) in the pores, and the concentration of solutes in the pore water (as described e.g. by the classical Archie's law, 1942). Note, however, that other factors may play a role, such as clay content (Rhoades et al., 1976; Waxman and Smits, 1968) and temperature (e.g., Campbell et al., 1949). However, in general, it is possible to estimate water content changes from changes in electrical resistivity over time (and space) provided that pore water salinity does not vary dramatically. While ERT has been attempted for quantifying root biomass on herbaceous plants (e.g. Amato et al., 2009), the main use of this technique in this context aims at identifying changes in soil water content in space and evolution in time (e.g., Michot et al., 2003, 2016; Srayeddin and Doussan, 2009; Garré et al., 2011; Cassiani et al., 2012, Brillante et al. 2015). With specific reference to RWU, Cassiani et al. (2015, 2016), Consoli et al. (2017) and Vanella et al. (2018) used time-lapse ERT with 3D cross-hole configurations to monitor changes in soil electrical resistivity caused by irrigation and RWU for different crops (apple and citrus trees). It should also be noted that RWU and the release of different exudates by fine roots modify soil water content and resistivity at several temporal scales (York et al., 2016).

On the other hand, evidence suggests that roots themselves may produce signals in ERT surveys (Amato et al., 2008; Werban et al., 2008); however, these signals are often difficult to separate from soil heterogeneities and soil water content variations in space (Rao et al., 2019). Nevertheless, in most cases, the ranges of electrical resistivity of soil and roots overlap, and while the amplitude of contrasts varies according to the soil resistivity and tree species (e.g. Mary et al., 2016), the direct identification of root systems using ERT is often impractical.

Recently, the Mise-A-La-Masse (MALM) method has been proposed for plant root mapping. MALM is a classical electrical method (Parasnis, 1967) originally developed for mining exploration, but also used more recently e.g. in the context of landfill characterization (De Carlo et al., 2013) as well as conductive tracer test monitoring (Osiensky, 1997; Perri et al., 2018). In MALM, an electrical current is injected into a conductive body with a return current electrode far away ("at infinity"), and the resulting voltage is measured at the ground surface or in boreholes, again with a reference electrode at infinity: the shape of voltage contour lines is informative about the extent and orientation of the conductive body. This idea can be applied to the plant stem and roots system, considering that electrical current can be transmitted through the xylem and phloem (on either side of the cambium), where sap flow takes place. The main assumption is that fine root connections and mycorrhiza at the contact between roots and soil convey the injected current into the soil where this contact is efficient, thus appearing as a distribution of current sources in the ground. The location of these sources should correspond to the locations of active contacts between roots and soil, and could be identified starting from the measured voltage distribution at the ground surface or in boreholes. This approach has been recently tested by Mary et al. (2018, 2019) on vine trees and citrus trees, showing that current injection in the stem and in the soil just next to the stem produces very different voltage patterns, thus confirming that the stem-roots system conveys current differently from a direct injection in the ground.

In this study we present the results of an infiltration experiment conducted in a Bordeaux vineyard (France). This paper is meant to be an extension of Mary et al. (2018) and to focus on the results of an infiltration experiment. The experiment was monitored (also) using time-lapse 3D ERT and time-lapse MALM measurements, the latter performed by injecting current in the vine trees stems. This study had the following goals:

(a) define a non-invasive investigation protocol capable of "imaging" the root activity as well as the distribution of active roots, at least in terms of their continuum description mentioned above, under varying soil water content conditions;

(b) integrate the geophysical results with mass fluxes measurements in/out of the soil-plant continuum system using a simple 1D simulation reproducing the infiltration experiment.

(c) give recommendations for future experiments focusing on the method validation.

## 2 Methodology

### 2.1 Site description

The study was conducted in a commercial vineyard (Chateau La Louviere, Bordeaux) in the Pessac Leognan Appellation of France (long 44°44'15''N, lat 0°34'45''W). The climate of the region is oceanic with a mean annual air temperature of 13.7 °C and about 800 mm annual precipitation. Grapevine trees are planted at 1 m distance along the rows, and the rows are spaced about 1.5 m. We focused our interest on two neighbouring plants.

The vineyard is not irrigated. The soil is sandy down to 1 m depth with sandy clay below, down to 1.75 m, and calcareous at depth. Due to its larger particles and thus smaller surface area, the sandy layer has a relatively poor water retention capacity. Nevertheless, the water supply of the vine plant is not a limiting factor (refer to Fig. 2 and Mary et al. (2018) for more details about the plants and soil type). We concentrated our monitoring on only two neighbouring grapevines (Fig. 1), which differ in age and size: plant A was smaller and younger, plant B was considerably larger and older.

### 2.2 Meteorological measurements and irrigation schedule

Hourly meteorological data were acquired by an automatic weather station located about 300 m from the plot and managed by DEMETER (Agrometeorological Service - www.meteo-agriculture.eu/qui-sommes-nous/lhistoire-de-demeter). These micrometeorological data were valuable to estimate the initial soil conditions and the changes in time (Figure 2). Potential evapotranspiration (ETP) was computed according to the Penman-Monteith formula accounting for the incoming short-wave solar radiation, air temperature, air humidity, wind speed and rainfall measured by the station. Prior to June 19, 2017, date of the first field data acquisition, little precipitation was recorded for 5 days (only 2.5mm on June 13) and only 18mm cumulative precipitation was recorded during the entire month of June 2017. The mean air temperature was very high (35°C under a well-ventilated shelter). Consequently, the plants were probably suffering from water deficit at the time of the experiment. Thus, at the start of the experiment, we assumed that the soil water content (SWC) around the plants was probably below to field capacity. As shown in Figure 2, the evapotranspiration rate was about 5.6 mm/day.

The controlled infiltration experiment was conducted using a sprinkler installed between the two monitored plants, placed at an elevation of 1.4m, in order to apply irrigation water as uniformly as possible. The irrigation started on June 19, 2017 at 13h00 and ended two hours later (15h00) for a total of 260 litres (104 l/h). Runoff was observed due to topography and probably induced more water supply for plant A that is located downhill. The irrigation water had an electrical conductivity of 720μS/cm at 15°C.

### 2.3 ERT and MALM data acquisition

We carried out a time-lapse ERT acquisition, based on custom-made ERT boreholes (six of them, each with 12 electrodes), plus surface electrodes (Fig. A1). The six boreholes were placed to form two equal rectangles at the ground surface. Each rectangle size was 1 m by 1.2m respectively in the row and inter-row line directions, with a vine tree placed at the centre of each rectangle. The boreholes were installed in June 2015 and a good electrical contact with soil was already achieved at the time of installation. The topmost electrode in each hole was 0.1 m below ground, with vertical electrode spacing along each borehole equal to 0.1 m. In each rectangle, 24 surface stainless steel electrodes (14mm diameter), spaced 20 cm in both horizontal directions, surrounded the plant stem arranged in a five by five regular mesh (with one skipped electrode near the stem). Note that after testing smaller electrode size in surface, we finally adopted larger ones since they ensured a better contact in the loose soil and were heavier and more firmly grounded (3cm out of 12) to resist irrigation. We conducted the acquisitions on each rectangle independently. Each acquisition was therefore performed using 72 electrodes (24 surface and 48 electrodes in 4 boreholes) using an IRIS Syscal Pro resistivity meter. For all measurements we used a skip 2 dipole-dipole acquisition

(i.e., a configuration where the current dipoles and potential dipoles are three times larger than the minimal electrode spacing).
The total dataset includes three types of measurements: 430 surface-to-surface, 2654 surface-to-borehole and 4026 in-hole
measurements.
In addition to acquiring ERT data, we also acquired MALM data. MALM acquisition was logistically the same as ERT and
was supported by the same device, but used a pole-pole scheme (with two remote electrodes). Borehole and surface electrodes
composing the measurement setup were used as potential electrodes, while current electrode C1 was planted directly into the
stem, 10 cm from the soil surface, with an insertion depth of about 2 cm, in order to inject current directly into the cambium
layer. The two remote electrodes C2 (for current) and P2 (for voltage) were placed approximatively at 30m distance from the
plot, in opposite directions. Note that for MALM (unlike than for ERT), one corner surface electrode was put near the stem in
order to refine the information at the centre of each rectangle.
Each MALM acquisition was accompanied by a companion MALM acquisition where the current electrode C1 was placed
directly in the soil next to the stem rather than in the stem itself. In this way the effect of the plant stem-root system in conveying
current can be evidenced directly comparing the resulting voltage patterns resulting from the two MALM configurations.
For both ERT and MALM, we acquired both direct and reciprocal configurations (that swap current and voltage electrode
pairs), in order to assess the reciprocal error as an estimate of measurement error (see e.g. Cassiani et al., 2006). Note that for
the MALM case, reciprocals may not be the best solutions to estimate data quality as it has been shown in Mary et al. (2018),
possibly because of non-linearity caused by current injection in the stem.
We adopted a time-lapse approach, conducting repeated ERT and MALM acquisitions over time in order to assess the evolution
of the system's dynamics under changing moisture conditions associated with the infiltration experiment. We conducted
repeated measurements starting on 19 June 2017 at 10:20 LT, and ending the next day at about 17:00 LT. The schedule of the
acquisitions and the irrigation times is reported in Table 1.
**2.4 Forward hydrological model and comparison with geophysical results**
Hydrus 1D (Simunek, J. et al., 1998) was used to simulate cumulative infiltration and water content distributions for plant B
(the larger one). The results from geophysical data acquisition were used to feed the hydrological model initial conditions.
Boundary conditions were set for the column respectively as an atmospheric BC with surface run off (observed during the
experiment) and triggered irrigation for the upper part, and free drainage for the lower part (see Figure 2). We assumed that
the retention and hydraulic conductivity functions can be represented by the Mualem-van Genuchten model (MVG, Mualem,
1976; van Genuchten, 1980). Soil hydraulic parameters were directly inferred using grain size distribution and the pedo-
transfer functions from the Rosetta software (Schaap et al., 2001). From the pit information (Mary et al., 2018), we assumed a
uniform soil type along a 1D column ranging from 0 to 1.2m depth (Figure 2c). We used two types of time variable boundary
conditions: (i) the irrigation rate changing with time, which was measured during the course of the experiment, and (ii) the
potential evapotranspiration estimated according to meteorological data. We neglected direct evaporation. The root profile has
been inferred from the MALM result at background (pre-irrigation) time using the average value along horizontal planes
(Figure 2b) discretised every 20cm. We used the functional form of RWU proposed by Feddes et al. (1978) with no water
stress compensation and a non-uniform root profile between 0 and 0.7 m depth.
The link between the forward hydrological and the geophysical model is a petrophysical relation which transforms electrical
resistivity distributions into the corresponding simulated water content ($\theta_{ERT}$) distributions. There are several petrophysical
models of varying complexity to relate water content with electrical resistivity (e.g. Archie, 1942; Waxman and Smits, 1968;
Rhoades et al., 1976; Mualem and Friedman, 1991). We adopted Archie's approach with the following parameters: pore water
conductivity was assumed equal to the electrical conductivity of the water used for the irrigation (720 µS/cm) for all the time
steps. The porosity was assumed to be equal to the soil saturated water content ($\theta_s$), the cementation factor (m) equal to 1.3
and the saturation exponent (n) equal to 1 (typical values notably described in Werban et al., 2008). We considered
homogenous soil distribution, so only one petrophysical relationship was necessary. Initial water content was inferred after
transformation and reduction by averaging to 1D the ER values obtained during background time $T_0$. We obtained a non-
homogeneous initial water content for the hydrological simulation varying from 0.1 to 0.27 cm3.cm-3 (Fig. 2a). In order to
compare the model results with the geophysical data, we used control points at 0, 0.2, 0.4, 0.6, 0.8m depth.
**2.5 Data analysis and processing**
*2.5.1 Micro-ERT time lapse analysis*
The inversion of ERT data was conducted using the classical Occam's approach (Binley and Kemna, 2005). We conducted
both absolute inversions and time-lapse resistivity inversions, as done in other papers (e.g. Cassiani et al., 2015, 2016). We
used for inversion only the data that pass the 10% reciprocal error criterion at all measurement times. A large percentage of
the data had reciprocity errors below this threshold. We inverted the data using the R3t code (Binley, 2019) adopting a 3-D
mesh with very fine discretization between the boreholes, while larger elements were used for the outer zone. Most of the
inversions converged after fewer than 5 iterations, and the final RMS errors respect the set convergence criteria (Table 1). For
the time lapse inversion, we followed the procedure described e.g. in Cassiani et al. (2006) in order to get rid of systematic
errors and highlight changes in term of percentage of ER ratios compare to the background time. Time-lapse inversions were
run at a lower error level (consistently with the literature – e.g. Cassiani et al., 2006) equal to 5%. At this threshold 65% (in
mean) of the data passed the reciprocity. A total number of 687 points were used during the inversion after selection of common
set between all-time steps.
*2.5.2 MALM modelling and source inversion*
The MALM processing applied to a plant is thoroughly described in Mary et al (2018). Here we only recall the mathematical
background on which the method relies on and some advances compare to the previous approach described by Mary et al.

226 (2018).

In MALM, we measure the voltage $V$ (with respect to the remote electrode) at N points, corresponding to the N electrodes
locations, $x_1, x_2, …, x_N$. Voltage depends on the density of current sources $C$ according to Poisson's equation:

$$\nabla \cdot (\sigma \nabla V) = C \quad , \tag{1}$$

where $\sigma$ is the conductivity of the medium, here assumed to be defined by the conductivity distribution obtained from ERT
data inversion. The main idea behind the source inversion is to identify the distribution of $M$ current sources $C(x,y,z)$ – in
practice located at the mesh nodes $C=[C_1, C_2, …, C_M]$ – that produce the measured voltage $V$ distribution in space. Given a
distribution of current sources, and once $\sigma(x,y,z)$ is known from ERT inversion, the forward problem is uniquely defined and
consists in the calculation of the resulting $V$ field. Conversely, the identification of $C(x,y,z)$ distribution given $V(x,y,z)$ and
$\sigma(x,y,z)$ is an ill-posed problem, that requires regularization and/or a priori assumptions in order to deliver stable results.
Different approaches are possible – for a detailed analysis in this context see Mary et al. (2018). In this paper we have used
the simplest approach, i.e. we assumed that one single current source was responsible for the entire voltage distribution. For
each candidate location the sum of squares between computed and measured voltages was used as an index of misfit of that
location as a possible MALM current source in the ground. Mary et al. (2018) introduced a simple index that can be mapped
in the three-dimensional soil space and that measures the misfit that a specific location is the (single) current source generating
the observed voltage field. This index ($F_I$) is defined as:

$$F_{1,i}(\mathbf{d}_m, \mathbf{d}_{f,i}) = \left\| \mathbf{d}_m - \mathbf{d}_{f,i} \right\|_2^2 \quad , \tag{2}$$

where $\mathbf{d}_m$ is a vector of measured voltage (normalised), and $\mathbf{d}_{f,i}$ is a vector of modelled voltage corresponding to a single
source injecting the entire known injected current at the i-th node in the mesh. The forward modelling producing the $\mathbf{d}_{f,i}$ values
is based on the direct solution of the DC current flow in a heterogeneous medium, such as implement in the R3t Finite Element
code (Binley, 2019). Thus, the $F_1$ inversion accounts naturally for the heterogeneous electrical resistivity of the 3D soil volume,
also in its evolution over time (e.g. as an effect of irrigation and RWU).
A more advanced objective function, which considers the presence of distributed sources, has also been introduced by Mary
et al. (2018). Here we propose several important changes to that approach, on the basis of the work by Peruzzo et al. 2019 who
proposed a linearized form of the problem. In this case, the cost function $F_2$ consists of error-weighted data misfit $\Phi_d$ and
model roughness $\Phi_m$ containing model relative smallness and smoothness both weighted by the regularization parameter $\lambda$:
$$F_2 = \Phi_d(\mathbf{m}) + \lambda\Phi_m(\mathbf{m}) = \|\mathbf{W}_\varepsilon(\mathbf{d}_m - f(\mathbf{m}))\|_2^2 + \lambda(\|\mathbf{W}_s(\mathbf{m} - \mathbf{m}_0)\|_2^2) . \qquad (3)$$


Given a set of N voltage measurements, minimization of the objective function, $F_2$, given by Eq. (3), produces a vector of M
current sources densities $\mathbf{c_j}$ ( j = 1,2,…,M), where $\mathbf{d}_m$ is the data vector, f($\mathbf{m}$) is the forward model that relates the model $\mathbf{m}$ to
the resistances, $\mathbf{W}_s$ is a smoothness operator, $\mathbf{W}_\varepsilon$ is an error weighting matrix, and $\lambda$ is a regularization parameter that
determines the amount of smoothing imposed on $m$ during the inversion. An L-curve analysis is used to identify the optimal
regularisation parameter $\lambda$. In the revised algorithm all candidate current sources are kept during the inversion. Thus, there is
no more a need to identify a threshold for which some sources are rejected. However, the misfit of $F_1$ is transformed into a
normalized initial model ($\mathbf{m_0}$) of current density via the inverse ($1/F_1$) transformation. During the inversion of the current
density, we adopted a relative smallness regularisation as a prior criterion for the inversion i.e. the algorithm minimizes ||m -
$m_0$ ||², where $m_0$ is a reference model to which we believe the physical property distribution should be close. Lastly, current
conservation was respected since the sum of $\mathbf{c_j}$ was equal to 1 at the end of the inversion iterations.
**3 Results**
**3.1 Background, irrigation time and monitoring of ERT measured data**
The soil electrical conductivity during the period prior to irrigation (see ERT results in Figure 2a and 3b, respectively for plants
A and B) ranged from 50 to 200 $\Omega$m, with a median value around 100 $\Omega$m, a range that is reasonable for a dry sandy soil. For
plant A, the smaller plant, the highest resistivity values were distributed at about 0.5 m depth (Figure 2a). For the larger plant
B (Figure 2), the positive resistivity anomalies are more diffused and less resistive (150 $\Omega$.m) compared to plant A, which
reach larger depths. The very small-scale anomalies observed at the soil surface are likely to be caused by heterogeneous direct
evaporation patterns or different soil compaction. The background time ($T_0$) for both plants revealed a low resistive layer
ranging in depth from 0 to 0.35 m for plant A and from 0 to 0.25 m for plant B. More interesting are the resistive anomalies at
intermediate depths. As observed in other case studies (e.g. Cassiani et al., 2015, 2016, Consoli et al., 2017; Vanella et al.
2018), these higher resistivity values are likely to be linked to soil saturation decrease caused by RWU, particularly in
consideration of its intensity during this time of the year (June) for non-irrigated crops. Of course, we cannot fully exclude that
higher resistivity is also related to woody roots presence, especially when they are dense. Besides, roots could also have
induced soil swelling creating voids acting like resistive heterogeneities.
The $T_1$ time step was collected during the irrigation, at 2h for plant A and at 30 minutes for plant B after the beginning of the
irrigation, so the variations of ER values are not directly comparable for the two plants. Figure 4a shows the resistivity
distribution during irrigation (at time step $T_1$) and after irrigation ($T_2$ to $T_5$) for plant B. The input of low resistivity water
(13.88 $\Omega$m, measured in laboratory) caused a homogeneous drop of the resistivity values (as much as 100 $\Omega$m difference)
around plant B. The observed resistivity decrease in the upper 40 cm can be attributed to the presence of a porous layer, and
correspondingly fast infiltration. A similar drop can be seen for the plant A (Fig. B1). This is an indirect evidence that water
infiltrated in both areas (that are next to each other) with no difference in soil hydraulic properties. For the time after irrigation,
it is difficult to appreciate the change in resistivity from the absolute values while time-lapse inversion (Fig. 4b) shows that
the main increase in ER (up to 140% of the background value), was located in the upper layers (< 0.3m depth) and occurred
between the background time and $T_3$. Note that the acquisition time $T_3$ corresponds to the morning of the following day, since
no measurement were taken overnight, and the acquisition time match with the start of the increase of ET and mean air
temperature. No increase was observed on plant A (Fig. B1). After $T_3$, no positive change in ER was observed.

## 3.2 Background and irrigation time steps of MALM measured data

Figure 5 shows the raw results of MALM acquisition on plant B, during background and irrigation, for both soil and stem
injection configurations. Note that voltages are normalized against the corresponding injected current. For both surface and
borehole electrodes the normalized voltage distribution can be compared against the one expected from the solution for a single
current electrode, idealized as a point injection of current $I$ at the surface of a homogeneous soil of resistivity $\rho$:

$$V = \frac{I\rho}{2\pi r} \qquad , \qquad (4)$$

where $r$ is the distance between the (surface) injection point and the point where voltage $V$ is computed (see Fig.5e for a
comparison). In all cases, both for surface and borehole electrodes, and both for stem and soil current injection, the resistance
patterns are deformed with respect to the solution of Eq. (4) for a homogeneous soil. Some pieces of evidence are apparent
from the raw data already:
a. In all cases, the pattern of surface and subsurface resistance is asymmetric with respect to the injection point (in the

stem or close to it, in the soil) and thus different from the predictions of Eq. (3); this indicates that current pathways

are controlled by the soil heterogeneous structure: note that at all times there is a clear indication that a conductive

pathway extends from the plant to the right-upper corner of the image (this would be the classical use of MALM –

identifying the shape of conductive bodies underground). Note that spatial variations of resistance between boreholes

are consistent with surface observations i.e. the maximum resistance was measured on the borehole 4 located in the

top right corner of the plot;

b. The resistance patterns in the case of stem injection are clearly different from the corresponding ones obtained from

soil injection. In particular, injecting in the soil directly produces a stronger resistance signal both at the surface and

in the boreholes than the corresponding resistance in the case of stem injection: this difference clearly points towards

the fact that the plant-roots system must convey the current in a different way than the soil alone; tentatively the

observed resistance features would indicate a deeper current injection in the case of stem injection. Looking at the

qualitative differences between soil and stem injection in the borehole electrode data, the impact is very small at

depths larger than 0.6m;

c. For both soil and stem injection, local anomalies observed in the background image are either removed or smoothed

during the irrigation steps. The effect is equally pronounced in soil and stem injection, showing that this is caused

essentially by the change in resistivity induced by the change in soil water content (see Fig.5).

Similar features are observed for plant A (results shown in appendix C1 and C2). The full-time monitoring is also shown only
in appendix since a consistent and quantitative interpretation is not straightforward by a visual inspection of the raw MALM
data.

## 3.3 Inversion of virtual current sources to estimate roots extents

Figure 6 shows the iso-surfaces of fitness index (or misfit) $F_1$ (Eq. 2) for the background (pre-irrigation) conditions of plant B
(plant A in appendix C3) and for current injection in the soil and in the stem at all-time steps listed in Table 1. In all cases,
Figure 6 shows the iso-surface corresponding to the value $F_1$= 7V corresponding to the 25% misfit index (value selected after

analysing the evolution of the L-curve of sorted misfit $F_1$. The same threshold is fixed for all the time steps thus the images provide comparable information for all cases. Note, nevertheless, that the position of the active roots from one acquisition to the other during the irrigation experiment (or for different seasons) may vary, so the distribution of the misfit and ultimately the depth of the iso-surface describing active roots.

In particular, the $F_1$ procedure highlights the remarkable difference, for both plants A and B, between the injection in the stem and in the soil. Current injection in the soil produces a voltage distribution that, albeit corresponding to a heterogeneous resistivity distribution and thus different from the predictions of a simpler model such as Eq. (3), collapses effectively to one point, i.e. the point where current was effectively injected in the ground. On the contrary, when current is injected in the stem, the region of possible source locations in the ground is much wider, and depicts a volume that is likely to correspond to the contact points between roots and soil, i.e. the volume where roots have an active role in the soil especially in terms of RWU. While this latter interpretation remains somewhat speculative, at least in the present experimental context, nevertheless the different results between soil and stem injection can only find an explanation in the role of roots and their spatial structure.

The most interesting feature shown by Fig. 6 is that the likely source volumes do not change with time during irrigation except for the irrigation time $T_1$ for which the iso-surface extended slightly more at depth. Note that the $F_1$ procedure makes use of the changing electrical resistivity distributions caused by infiltrating water (see Fig.4) thus the result is not obvious, and indicates an underlying mechanism that is likely to be linked to the permanence of the roots structure over such a short time lapse.

Figure 7 shows the spatial distribution of the current density as an outcome of the minimisation of the $F_2$ function. Very similar observations to $F_1$ are driven from the current source density i.e. that current injection in the soil produces a current distribution collapses effectively to one point, i.e. the point where current was effectively injected in the ground, while when current is injected in the stem, the current distribution in the ground is much wider, and depicts a volume that is likely to correspond to the contact points between roots and soil. Note that the different time steps (Fig. C4) did not highlight changes in the distribution of current density suggesting that the region of RWU was relatively constant during the experiment.

**3.4 Electrical resistivity variations inside and outside the likely active roots zone**

Our assumption is that the region identified by MALM $F_1$ for the background time corresponds to the RWU region. The inner area (IN) is then defined as the area within the closed iso-surface at the background time $T_0$. As the changes in the estimated extent of the root zone are only minor (Fig. 6), it makes sense to evaluate the changes, as an effect of irrigation, in electrical resistivity within such stable estimated root zone. Figure 8 shows the ER variations of selected values in the zones outside and inside this estimated active root zone. It is apparent how irrigation causes a general decrease of electrical resistivity for both plants A (Fig. 8a) and B (Fig. 8b), and in both inner and outer regions. Note that even though the regions are different for the two plants, the behaviour is similar. Then at the end of irrigation we observe, for both plants, that resistivity continues to decrease outside the root active region, while it increases slightly inside. This behaviour is consistent with the fact that inside the region we expect that RWU progressively dries the soil, while outside this region resistivity continues to decrease (overall) as an effect (probably) of water redistribution in the unsaturated soil.

**3.5 One-dimensional simulation of the infiltration**

Figure 9a shows, the variations of the simulated soil water content ($\theta_{simu}$) with time for control points located at different depths (see Fig. 2 for the geometry) and Fig. 9b shows the comparison against the 3dimensional variations of ER transformed values to soil water content ($\theta_{ERT}$). Time steps of the ERT acquisition for starting time and end time are reported on Fig. 9a for an easier comparison with Fig. 9b. At $T_0$, values of soil water content are about 0.1, a value close to field capacity for this type of soil, as previously assumed (section 2.2) and in agreement with the literature. Despite all the assumptions and models'

limitation described later, the range of soil water seems also consistent between the simulation and the measured data. Note also that the dynamic is closely linked to the estimated ET and mean air temperature shown in Figure 2. The start and end time of the triggered irrigation are clearly identified respectively with a sharp increase following by a decrease of $\theta_{simu}$ at z=0, with a peak in SWC equal to 0.3. Between $T_1$ and $T_2$, only the upper surface (<0.2m depth) is affected by the irrigation front resulting in the increase of soil water content both visible in $\theta_{simu}$ and $\theta_{ERT}$ (Fig. 9b). The infiltration front reaches the depth of 0.4 m during the collection of ERT data at time $T_2$. Time $T_2$ marks the starts of a regular decrease of the soil water content overnight in the top 40cm soil. Time $T_3$, coincident with an increasing ET and mean air temperature, highlights a rupture from a slow decrease to a higher decrease rate particularly for the soil surface (the layer <0.2m depth), in agreement with the observed changes in $\theta_{ERT}$ (Fig. 9b). Overall, Fig. 9a and 9b show a good correlation between the dynamics of SWC changes predicted by the hydrological model ($\theta_{simu}$) and observed via the ER transformed values ($\theta_{ERT}$).

**4 Discussion**

The survey was carried out during a sunny summer season in a non-irrigated vineyard of the Bordeaux Region. The site is composed of sandy-loamy soil, thus there is a high infiltration rate during the experiment, and this would make it more difficult to distinguish RWU zones from infiltration zones as done for instance by Cassiani et al. (2015) using time-lapse ERT alone. The first objective of the study was to define a non-invasive investigation protocol capable of "imaging" the root activity as well as the distribution of active roots under varying soil water content. We demonstrated that the key additional information is provided by MALM which directly incorporates the ERT information in terms of changing electrical resistivity distribution in space including its evolution in time. MALM, and particularly its double application of current injection in the stem and in the soil next to it, uses electrical measurements in a totally different manner: here the plant-root system itself acts as a conductor, and the goal is to use the retrieved voltage distribution to infer where the current injected in the stem actually is conveyed into the soil: these locations are potentially the same locations where roots interact with the soil in terms of RWU. However, in order to try and locate the position of these points, it is necessary to know the soil electrical resistivity distribution at the time of measurements. At this scale of measurements, ERT provides 3D images of electrical resistivity distribution in the subsoil housing the root system. Fast acquisition allows the measurement of resistivity changes over time, which in turn can be linked to changes in SWC. This can be caused e.g. by water infiltration, or by RWU: in the latter case, negative SWC changes mapped through resistivity changes can be used to map the regions where roots exert an active suction and reduce SWC. However, water redistribution in the soil also plays a role in terms of resistivity changes. Thus some additional independent information about the location of active roots in the soil may help: this is the first coupling between ERT and MALM that has been integrated in the workflow. Considering the inverted MALM data as non-sensitive to soil water distribution has different potential useful impacts: the separation of contributions of root zone and outer area on ER values extracted from ERT help distinguish between soil processes such as RWU and hydraulic redistribution (hydraulic lift in particular).

Time-lapse ERT measurements gives clear evidence that injecting current in the stem and in the soil close to the stem produces different inversions even under changing soil water conditions. The soil injection produces a current density close to a punctual injection (located at the true single electrode location) whatever the soil water content. The stem injection helps identify a 3D region of likely distributed current injection locations, thus defining a region in the subsoil where RWU is likely to take place. The latter result is particularly useful, in perspective: when computing the time-lapse changes of electrical resistivity inside and outside this tentative RWU region during irrigation we clearly see that while inside resistivity increases (as an effect of RWU, as irrigation is still ongoing), outside resistivity decreases. Thus, our assumption that the region identified by MALM inversion (albeit very rough) corresponds to the RWU region is corroborated indirectly also by this evidence.

## 4.1 Comparison between geophysical data and hydrological model

A second objective of the study was to integrate the geophysical results in a simple 1D model of the infiltration experiment, that takes into account the observed water fluxes. Dupuy et al., (2010) advocated the use of roots systems described as "density" distributions. We assimilated the root distribution, derived the geophysical data, into the hydrological model. Attempts in this direction are very promising to describe the root functioning in the framework of continuum physics, i.e. the one endorsed by SPAC. The integration of modelling and data has proven a key component of this type of hydro-geophysical studies, allowing us to draw quantitative results of practical interest. For example, in our study it is apparent that although infiltration occurred during the peak of evapotranspiration (between 1pm and 3pm), very small RWU was observed before the second day. Nevertheless, after a certain time, RWU is observed while infiltration is still ongoing. Smaller RWU observed for the small plant A compared to plant B is also observed.

## 4.2 Recommendation for future experiments

In this field case study, we had very little available quantitative information that could allow the validation of the geophysical data in terms of the volume of soil affected by RWU. The final objective of this study was then to discuss issues for obtaining suitable validation data using existing methods and propose some recommendation for future experiments:

(i) Traditional root sampling methods should be the first line of validation although they have numerous potential pitfalls. As roots are underground, and thus invisible in their space-time evolution, and are also fragile, especially in their fine structure, the monitoring of their structure and activity using destructive methods such as trenches or air spade presents various limitations. In such approaches, even in the best case where fine roots may be sufficiently preserved and described, it is impossible to know where the active roots actually are. Active roots may be located only in one part of the whole root system. Destructive methods may help to validate the confidence area determined by $F_1$ but are not appropriated methods to validate the $F_2$ inversion.

(ii) We recommend the use of traditional methods (such as Time Domain Reflectometry-TDR and tensiometers) for future studies. Though punctual, these data can greatly facilitate the data calibration and validation of geophysical methods.

Finally, more research needs to be conducted to understand how MALM can provide information to be correlated with the actual RWU and thus to the estimated transpiration. The study of complex root-soil interactions requires that high time resolution and extensive data are collected and processed. In order to quantitively evaluate RWU using the variations of ER, many more data instants per day must be acquired. In this study, we only used ERT and MALM information to initialize the infiltration model, and only a qualitative comparison was conducted between model predictions and geophysical results. In the near future, a real assimilation scheme using data assimilation technique should be adopted.

## 5    Conclusions

This study presents an approach to define the extent of active roots distribution using non-invasive investigations, and thus particularly suitable to be applied under real field conditions. We applied a mix of ERT and MALM techniques, using the same electrode and surface electrode distribution. The power of the approach lies in the complementary capabilities of the two techniques in providing information concerning the root structure and activity. The approach has been tested in a vineyard during an irrigation experiment. Future experiments would require that high time resolution extensive data are collected, and the results are analysed in conjunction with data from traditional monitoring methods in order to qualitatively integrate geophysical results into a hydrological one. The presented approach can be easily replicated under a variety of conditions, as DC electrical methods such as ERT and MALM do not possess a spatial scaling per se, but their resolution depends on electrode

spacing as well as on other factors that are difficult to assess a priori, such as resistivity contrasts and signal to noise ratio.
Thus similar experiments can also be used in the laboratory, where more direct evidence of root distribution can be used to
further validate the method.
**Acknowledgements**
The authors wish to acknowledge support from the ERANET-MED project WASA ("Water Saving in Agriculture:
Technological developments for the sustainable management of limited water resources in the Mediterranean area"). The
authors from the University of Padua acknowledge support also from the University Research Project "Hydro-geophysical
monitoring and modelling for the Earth's Critical Zone" (CPDA147114). In addition, the information, data or work presented
herein was funded in part by the Department of Energy Advanced Research Projects Agency-Energy (ARPA-E) project under
work authorization number 16/CJ000/04/08 and Office of Science Biological and Environmental Research Watershed
Function SFA project under Contract Number DE-AC02-05CH11231. The views and opinions of authors expressed herein do
not necessarily state or reflect those of the United States Government or any agency thereof. Luca Peruzzo and Myriam
Schmutz gratefully acknowledge the financial support from IDEX (Initiative D'EXellence, France), the European regional
development fund Interreg Sudoe – Soil Take Care, no. SOE1/P4/F0023 – Sol Precaire.
**Data availability**
Data used to generate the figures can be accessed on the Padua Research Archive (Link to come after decision).

**Authors contributions**
GC, YW, and SH worked on the conceptualization of the research. BM curated the data. LP, BM, NC and JB collected the
data. BM prepared the formal analysis, designed and wrote the scripts for carrying out the simulation and inversion, and ran
the obtained the results. MS and GC supervised the field work. All the authors discussed the results. BM prepared the paper
with contributions from all the authors.
**Competing interests**
The authors declare that they have no conflict of interest.

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

**Table 1: schedule of the acquisitions and the irrigation times; Plant A and B are measured consecutively and consist each time of**
**three measurements: ERT, MALM stem and MALM soil. Assessment of data and inversion quality from the two last columns i.e.**
**respectively the percentage of data that passed the reciprocity (analysis at 10%) and RMS error at the end of the inversion.**

| Acquisition no. | Plant | Starting time (LT) | Ending time (LT) | Irrigation | Date | % of data retained (10% reciprocals) | Final RMS (Ohm.m) |
|---|---|---|---|---|---|---|---|
| **0 (background)** | A | 10:20 | 11:00 | | | 79 | 1.15 |
| | B | 12:20 | 13:00 | | | 91 | 1.76 |
| **1 (Irrigation)** | A | 15:00 | 15:30 | 13h00 to 15h30, 104lh-1 For both plants | Day 1 (19 June 2017) | 50 | 1.54 |
| | B | 13:30 | 14:00 | | | 68 | 1.31 |
| **2** | A | 17:00 | 17:30 | | | 69 | 1.36 |
| | B | 18:00 | 18.45 | | | 57 | 1.50 |
| **3** | A | 10:30 | 11:00 | | | 59 | 1.72 |
| | B | 9:30 | 10:00 | | | 80 | 1.24 |
| **4** | A | 14:00 | 14:30 | | Day 2 | 72 | 1.38 |
| | B | 15:00 | 15:30 | | | 80 | 1.53 |
| **5** | A | 18:00 | 18:30 | | | 70 | 1.23 |
| | B | 17:00 | 17:30 | | | 78 | 1.28 |


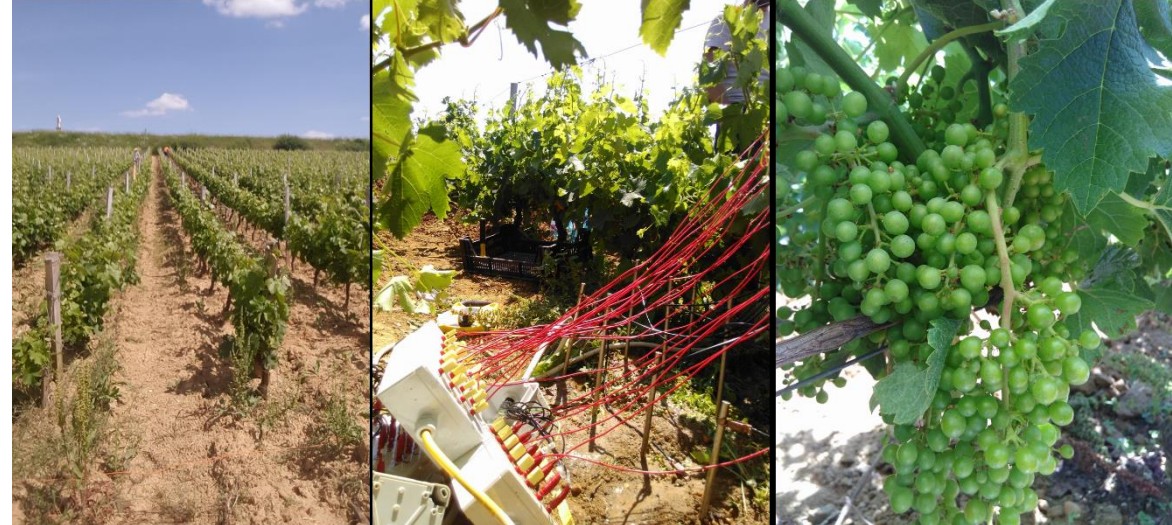

**Figure 1: picture of the field site in May 2017 (a) wired plants investigated (b) and grape status during the experiment in June**
**2017 (c)**

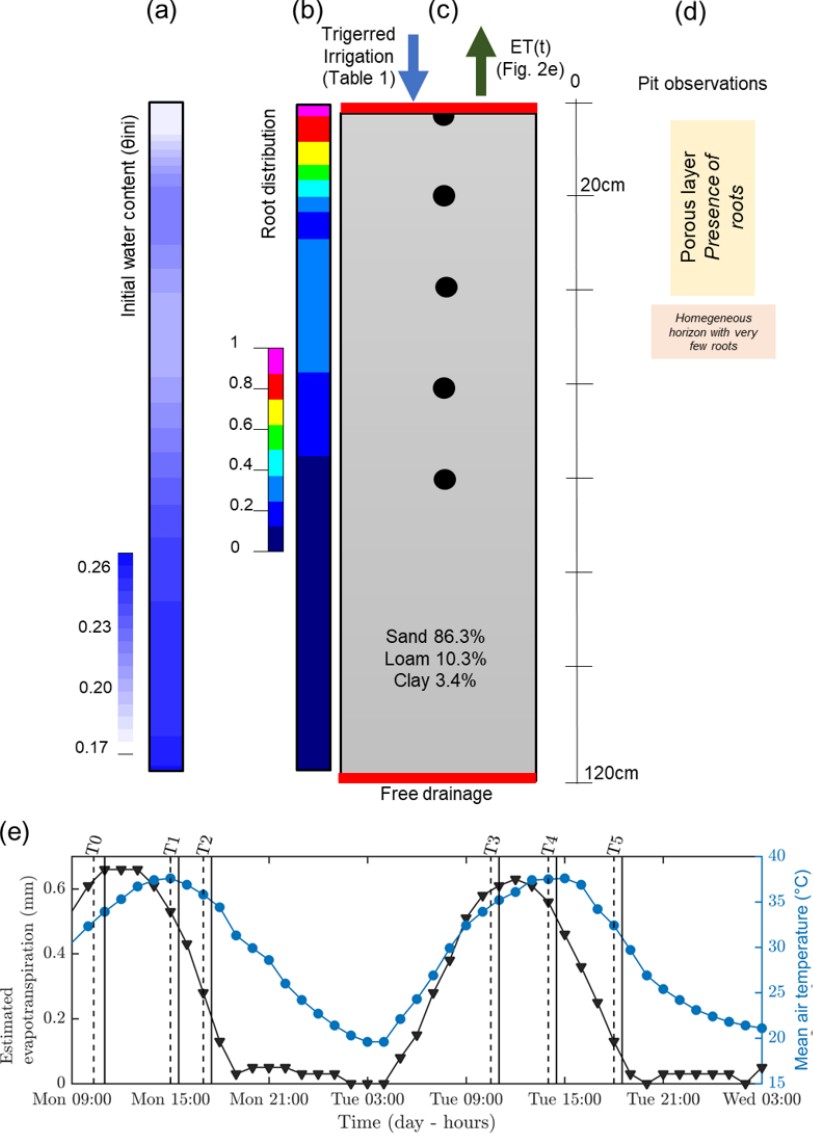


Figure 2: Initial (a,b,c) and time varying atmospheric conditions (e) used the hydrological simulation (e). From left to right (a-d), initial conditions on soil water content $\theta_{ini}$, root density (1/cm), soil type, and pit observations. (e) variation of temperature (blue line) and estimated evapotranspiration (black line) derived from a nearby meteorological station. The vertical lines indicate acquisition times for plant A (dashed and plain line respectively for the start and the end of the measurement, see Table 1).



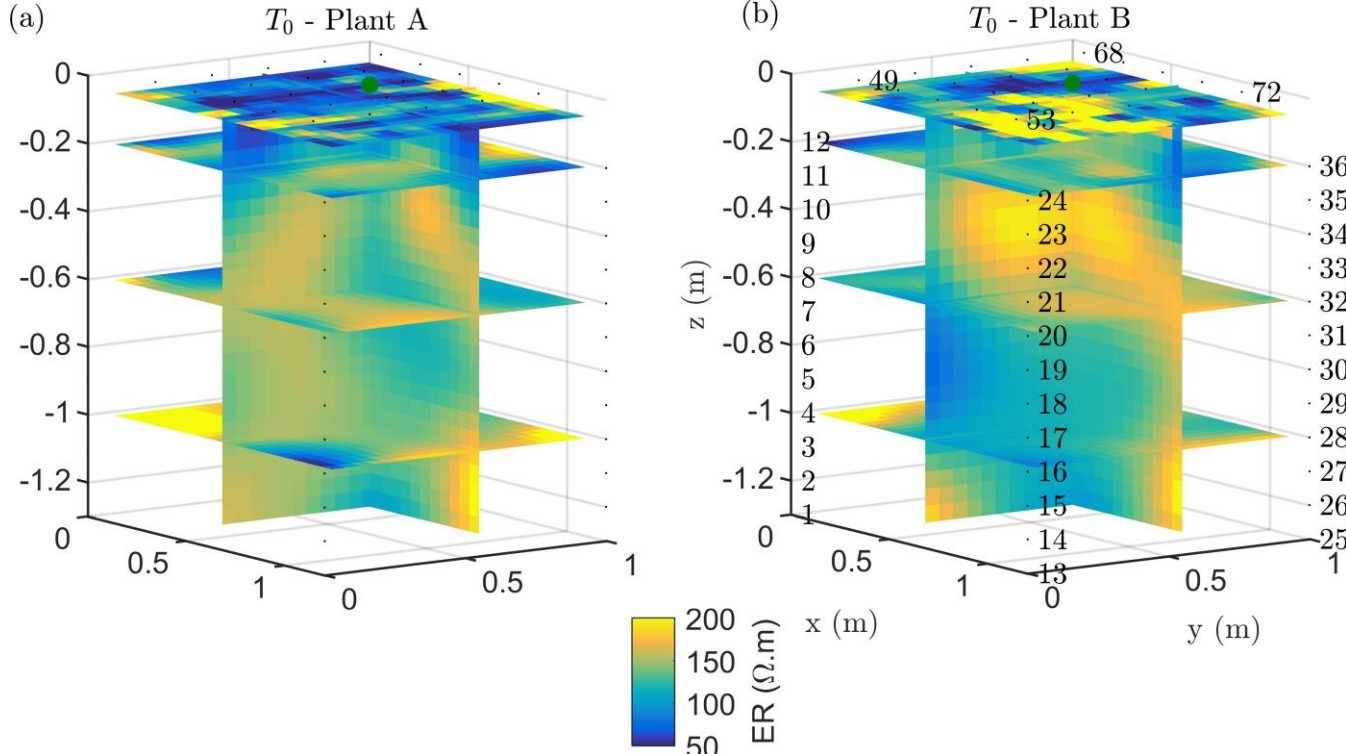


**Figure 3: Results of the 3-D ERT inversion for the background time T0 for plant A (b) and B (b). 3-D resistivity volume (log scale) sliced at the tree stem position (vertically) and at four depths (0.05, 0.2 0.6 and 1m), with the green point showing the location of the plant stem.**


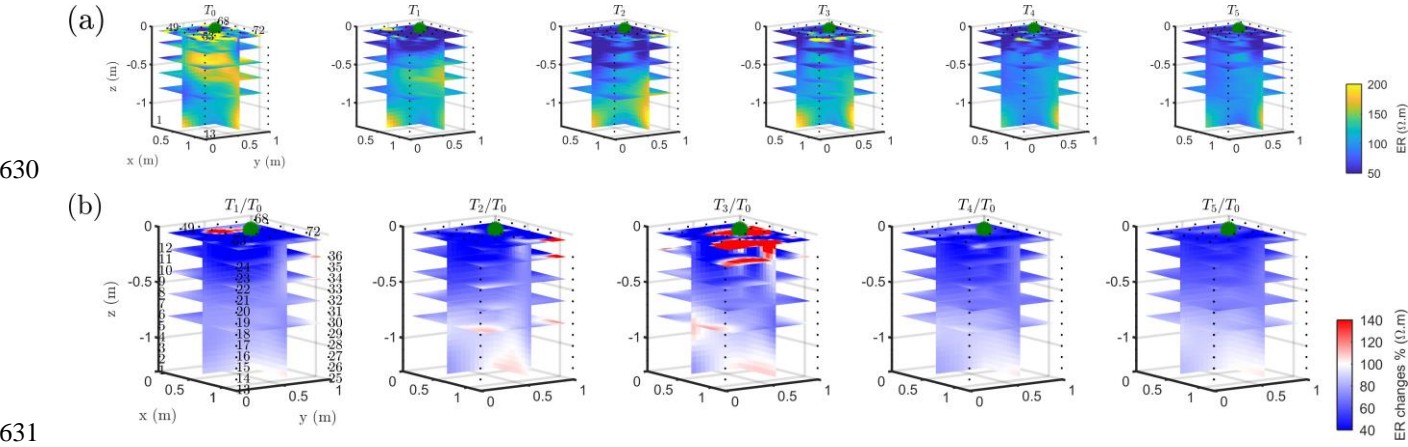

**Figure 4: 3D ERT results for plant B (plant A, in appendix Fig. B1). The volume is sliced at the tree stem position (vertically) and at five depths (0.05, 0.2, 0,4, 0.6 and 0.8 m). (a) 3D inversion of the resistivity (in Ωm, log scale) from the background time $T_0$, during irrigation $T_1$ and after irrigation. (b) time-lapse inversion (following Cassiani et al., 2006) showing the ratios (in % of ER changes) between time step $T_i$ and background time $T_0$ (100% in white means no change).**

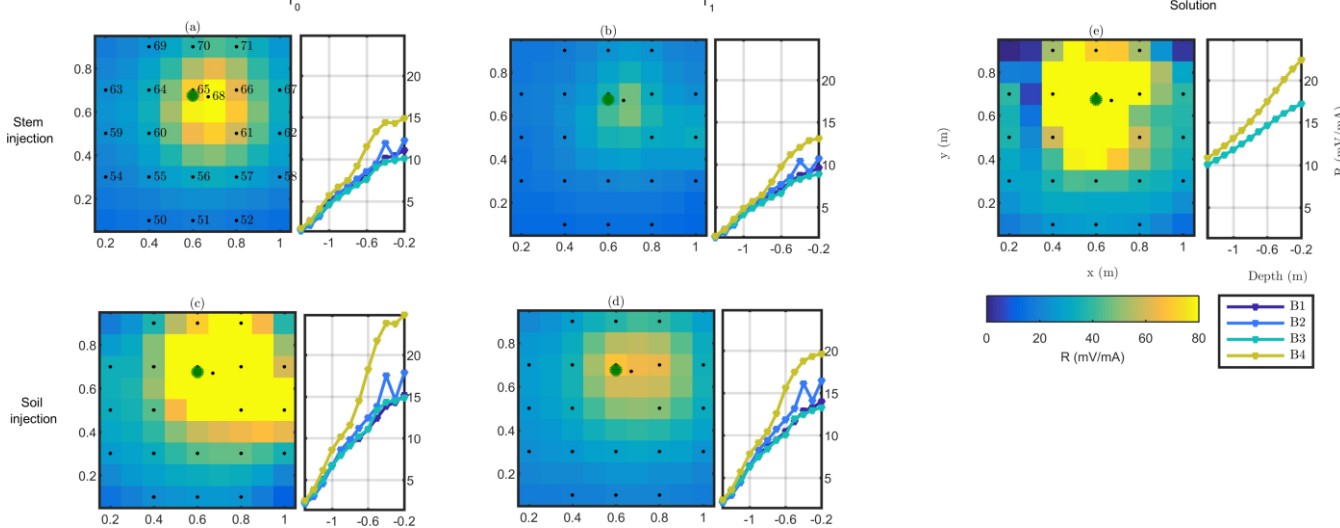

Figure 5: plant B~~A~~, MALM results showing variations in surface (horizontal plan) of resistance R (in mV/mA) for the initial state background $T_0$ (a,c) and irrigation $T_1$ (b,d) time steps. Comparison between the stem injection (a,b) and soil injection (c,d). The black points show the surface electrodes location. The green point shows the positions of the plant stem. Data are filtered using a threshold on reciprocal acquisition of 20%. (e) shows the solution using Eq. (4) for a homogeneous soil of 100 Ohm.m; The resistance between boreholes B1/B3 and B2/B4 (see legend) are identical and cannot be distinguished graphically in the case of (e).

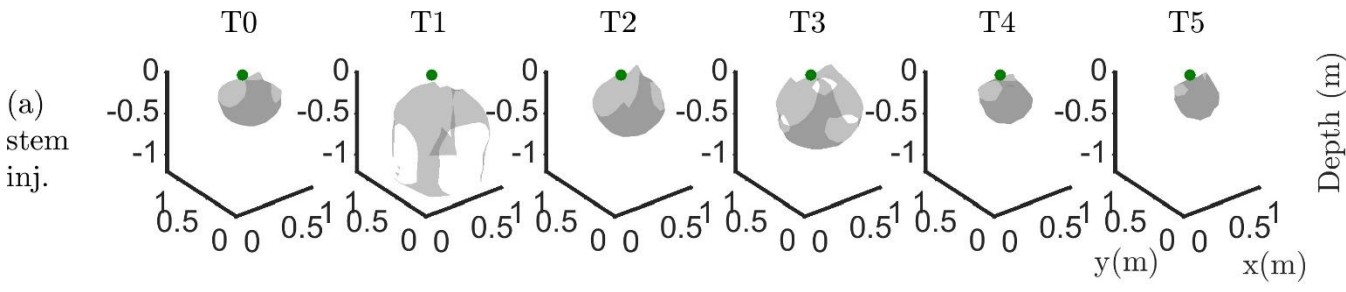

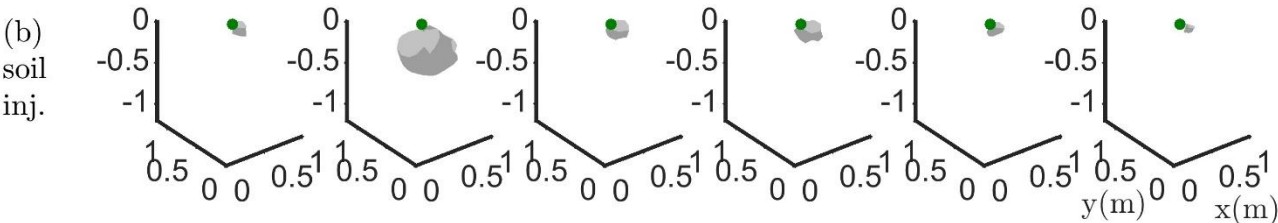


**Figure 6: iso-surface minimizing the $F_1$ function for plant B; during stem injection (a), during soil injection (b); Columns represent**
**the six times steps from $T_0$ to $T_5$. Green dot shows plant stem position. Threshold is defined by the misfit 25% of the normalised $F_1$**
**(value selected according to the evolution of the curve of sorted misfit $F_1$ and calculated for the tree injection at $T_0$ and kept constant**
**for all the time steps).**

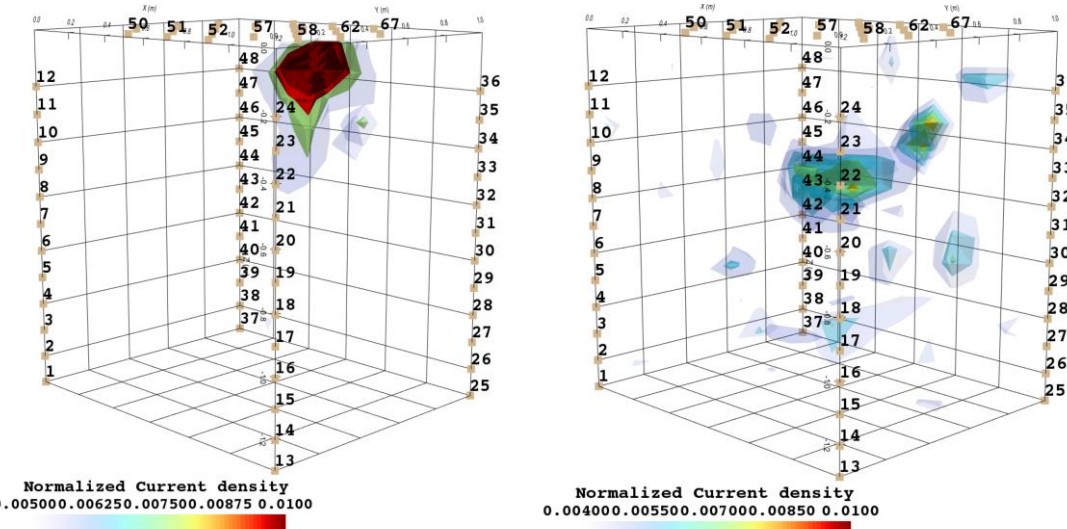


**Figure 7: current source density after minimization of the objective function $F_2$ as defined in Eq. (3). The results are relevant to the background time $T_0$ for the plant B, for the soil current injection (left) and the stem current injection (right).**




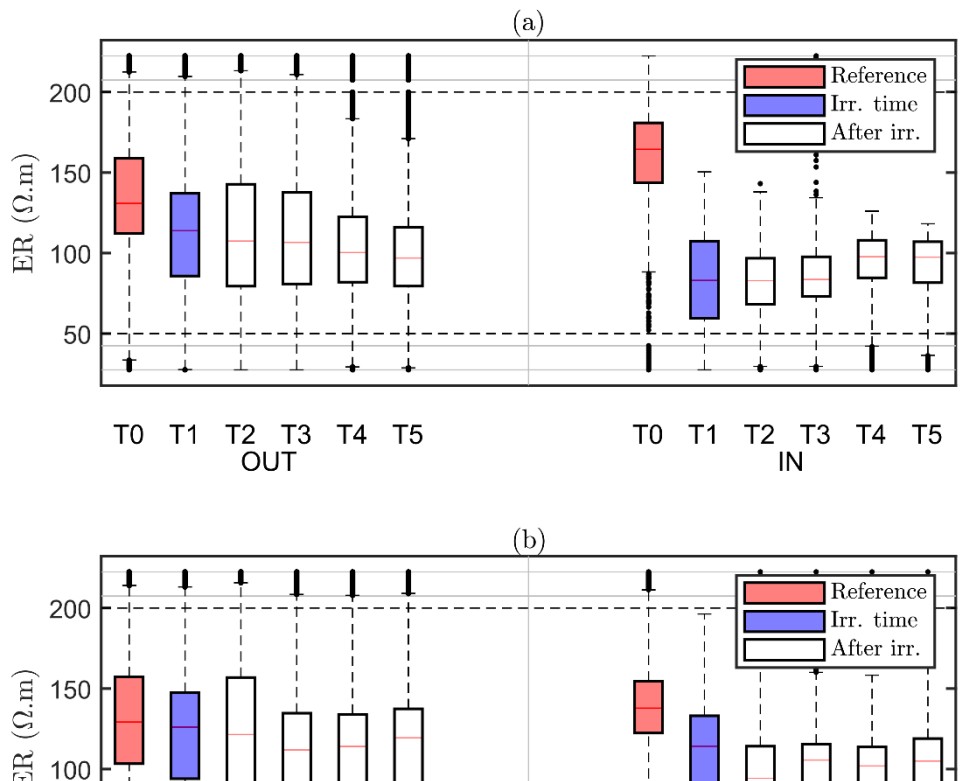


**Figure 8: boxplot distribution of ER time variations observed on the plant A (top) and plant B (bottom), for the values selected**
**outside (OUT, left part) and inside (IN, right part) of the region defined by the $F_1$ best fit sources (see Fig. 6a-$T_0$). The central mark**
**indicates the median, the bottom and top edges of the box indicated the 25th and 75th percentiles of ER data, respectively. The**
**whiskers extend to the most extreme data points (black dots) considered outliers. Each box corresponds to a given time step (see**
**table 1), indicated in the x-axis.**


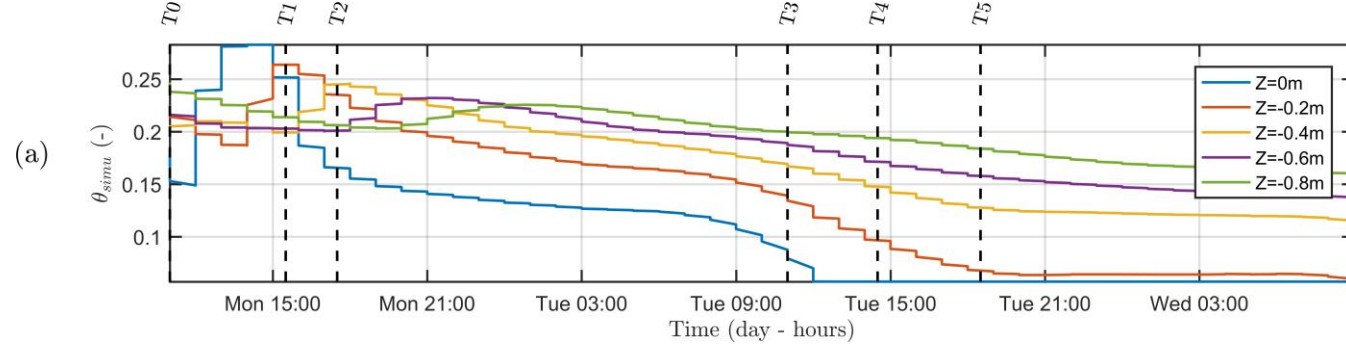


(a)

(b)

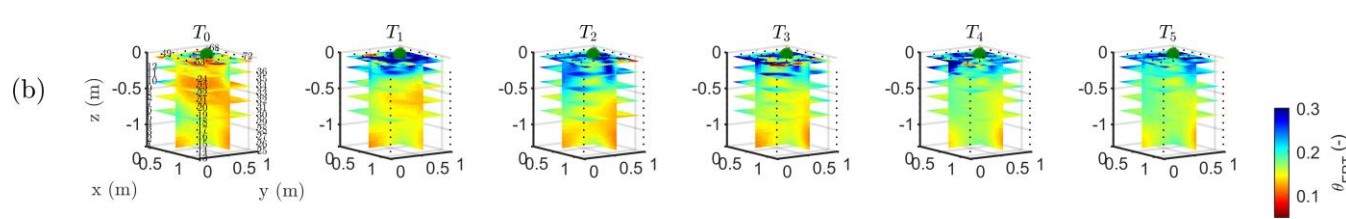

**Figure 9: (a) time variation of simulated soil water content ($\theta_{simu}$) at five depths. The vertical lines indicate the geophysical acquisition**
**times (dashed and plain line respectively for the start and the end of the measurement, see Table 1). (b) 3D variations of the ERT-**
**derived soil water content ($\theta_{ERT}$) for the time steps describe in table 1. Horizontal layer depths are identical to the control points of**
**the hydrological model.**

**Appendix A: set-up description**

Fig. A1: from left to right: 3D view of the surface (blue) and borehole (black) electrodes, view from the top and tranversal
view. Plant A was located downhill. Green dot shows plant stem positions.

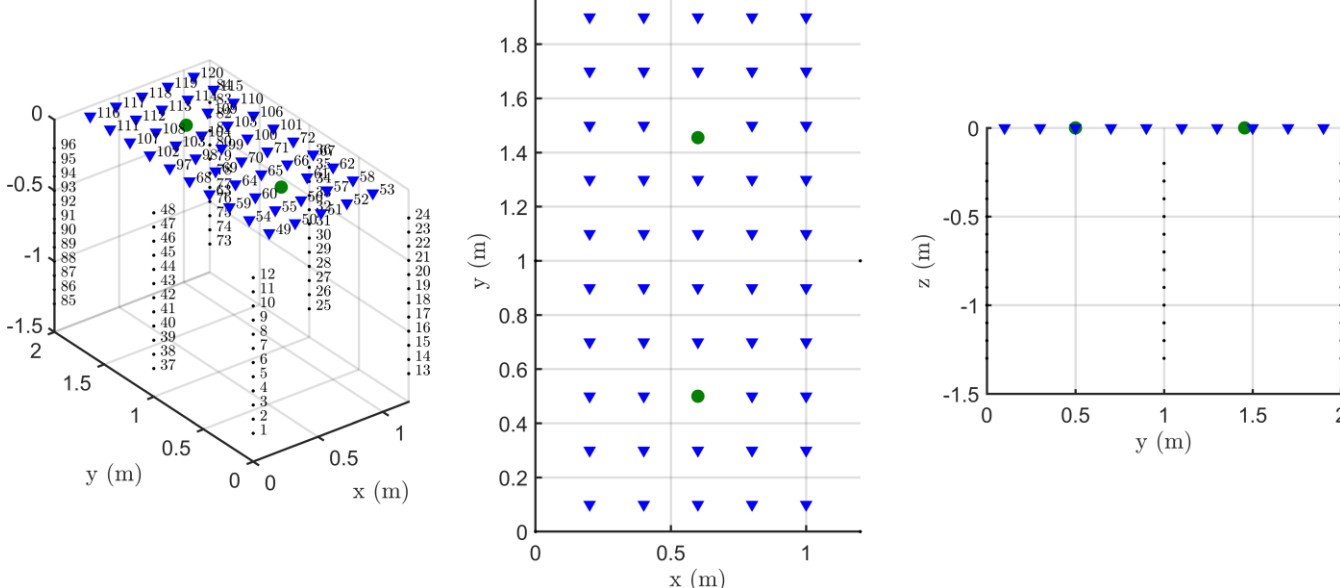



**Appendix B: ERT monitoring**

Fig. B1: 3D ERT results for plant A. The volume is sliced at the tree stem position (vertically) and at five depths (0.05, 0.2, 0,4, 0.6 and 0.8 m). (a) 3D inversion of the resistivity (in $\Omega$m, log scale) from the background time $T_0$, during irrigation $T_1$ and after irrigation. (b) time-lapse inversion (following Cassiani et al., 2006) showing the ratios (in % of ER changes) between time step $T_i$ and background time $T_0$ (100% in white means no change).

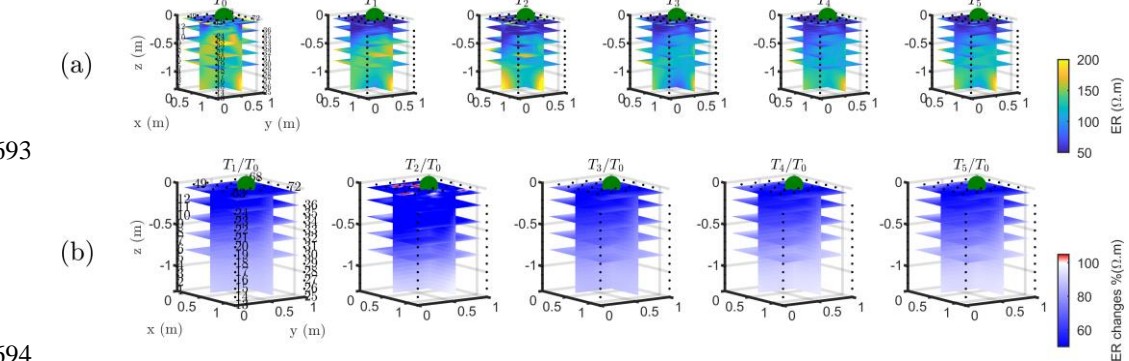

**Appendix C: MALM monitoring**

Fig. C1: Resistance distribution of the raw data of MALM time lapse monitoring for the plant B. First line results are relevant to the stem injection while second line refers to the soil control injection. Columns describe time evolution according to Table 1.

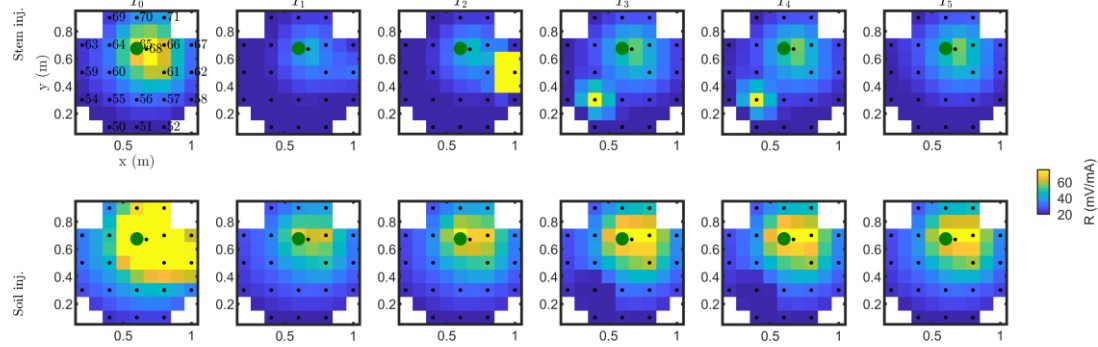

Fig. C2: Resistance distribution of the raw data of MALM time lapse monitoring for the plant A. First line results are relevant to the stem injection while second line refers to the soil control injection. Columns describe time evolution according to Table 1.

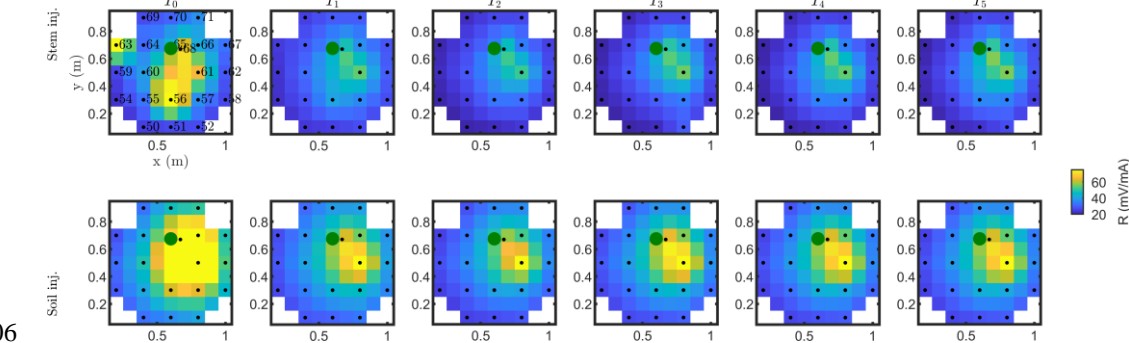

Fig. C3: iso-surface minimizing the F1 function for plant A; during stem injection (a), during soil injection (b); Columns
represent the six times steps from T0 to T5. Green dot shows plant stem position. Threshold is defined by the misfit 25% of
the normalised F1 (value selected according to the evolution of the curve of sorted misfit F1 and calculated for the tree injection
at T0 and kept constant for all the time steps).


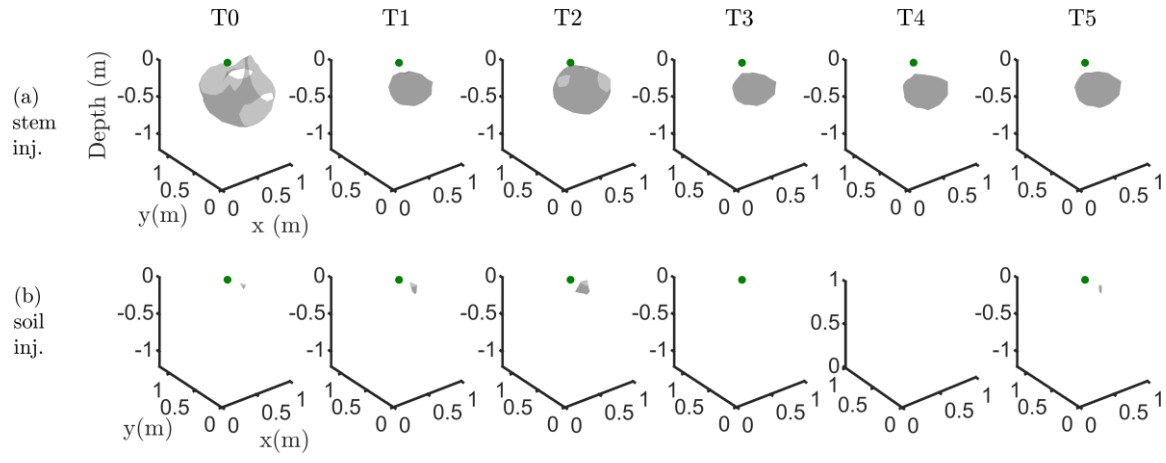


Fig. C4: Time-lapse evolution of the current source density after minimization of the objective function F2 as defined in Eq.
(3). The results are relevant to the background time T0 to T5 for the plant B, for the soil current injection on the left, and stem
current injection on the right.

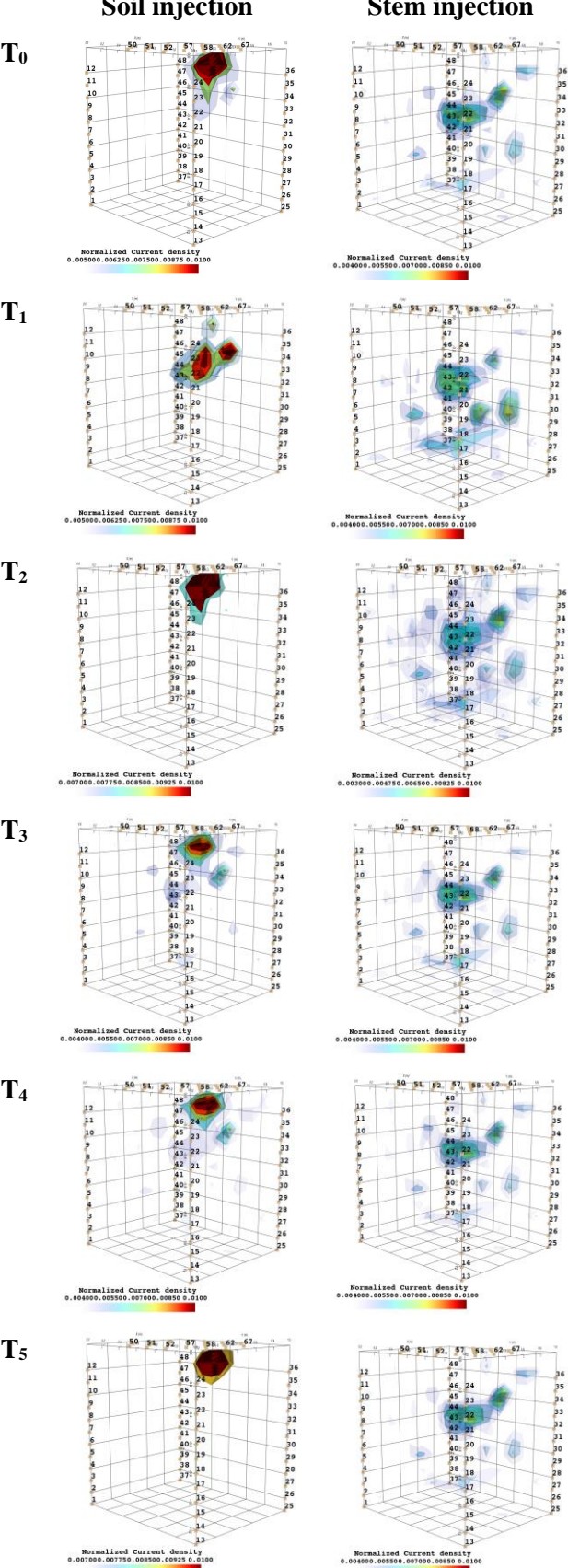
