# Peer review of "Time-lapse monitoring of root water uptake using electrical resistivity tomography and Mise-à-la-Masse: a vineyard infiltration experiment"

_SOIL, 2019_

## Referee Comment (RC1) · Anonymous Referee #1 · 7 Jun 2019

The manuscript "Time-lapse monitoring of root water uptake using electrical resistivity tomography and Mise-à-la-Masse: a vineyard infiltration experiment" by authors Mary et al investigates root characterization using the electrical MALM approach in a time-lapse setting.

As explained in the manuscript, new approaches to non-, or minimally invasive root characterization are urgently needed to increase our knowledge about the root zone, as well as to provide better data input for soil-plant-atmosphere (SPAC) modeling frameworks. As such I think the topic of the manuscript is relevant to the readership of SOIL and well worth investigating. However, in its current state I cannot support a publication of the text under review without major revisions. Some parts of the manuscript feel somewhat rushed, and perhaps some (or most?) of my issues can be solved by reformulations or some additional text?

——————— General Comments: ———————

My two major concerns are:

1) No validation data: The study uses neither independent information on the rooting depth or distribution, nor are soil information such as soil water content measurements, Archie-Parameters, or soil temperature data used to support the statements made (or used in the analysis of the data). The fact of missing validation data is also mentioned in the abstract, although not further discussed in the text.

In light of the novel, very promising, technical approach of MALM I don't think this should prevent the publication of the text, yet it should be actively discussed and conclusions should be limited to statements that can be made without validation data. Perhaps the text could be reworked to provide/develop recommendations for future experiments that deal with the problem of obtaining suitable validation data? As such, the direct predecessor paper, Mary et al 2018, and this study could be positioned as discussing the technical details of the approach, preparing future studies which focus on the validation aspect.

2) Overlap with Mary et al 2018: As far as I can see Mary et al 2018 and this study were conducted on the same field site and same plants within a few months time. As such they should be considered as companion studies. As far as I can see their stated objectives overlap massively (as do the conclusions):

Aims of Mary et al 2018 (page 5429):

"1. define a viable field protocol that uses jointly MALM and ERT to map active tree vine roots, 2. propose and analyze algorithms capable of identifying the location of active roots, and 3. test the algorithms above against real data from a French vineyard."
Aims of this study:

"(a) define a non-invasive investigation protocol capable of "imaging" the root activity as well as the distribution of active roots, at least in terms of their continuum description mentioned above; (b) Integrate the geophysical results with mass fluxes measurements in/out of the soil-plant continuum system."

My reading is that all aims of Mary et al 2018 are contained in aim a) of this study.

Adding to this, I was not able to find any information on the stated "integration of geophysical results with mass fluxes" (aim b) in the text, leaving only the duplicate aim a).

Also note that the time-lapse aspect is currently not discussed in detail (detailed below) in the text, the analysis of the time-lapse data does not take into account any dynamics such as daily evaporation information, and the conclusion mostly reflects the conclusions of Mary et al 2018, without significant conclusions regarding the application of MALM within a time-lapse context.

———————————————————- Specific comments/technical corrections: ————————————————

3) In addition, there are various (apparent?) inconsistencies between Mary et al 2018 and this text. However, this could be caused by the rather brief formulations in the text, not by actual errors. I suggest to rephrase.

3.1) Site description: Mary et al indicate sandy-clayey soil from 125-175cm, while this text puts this layer from 100-175m. I also wonder why the authors do not interpret the different information on root distribution given in Mary et al 2018. For example, Mary et al 2018 identifies the first soil layer as "with a first sandy horizon (0–40 cm depth), porous and soft." Looking at Figure 3 in this text, I wonder if the observed resistivity decrease in the upper 40 cm can be attributed to this porous layer, and correspondingly fast infiltration?

3.2) Mary et al 2018 state that active roots are located in the upper 0.3m (page 5436). This does not seem to be the case if the isosurfaces in Figure 6 (this text) are to be interpreted as root extension, although the same plants are measured. Note that this text also states, as one of the conclusions, that the MALM approach is relatively insensible to different water regimes, and thus I would expect similar results between both studies.

If this inconsistency is caused by slightly different interpretation of the term "active roots", I suggest to rephrase accordingly.

3.3) Mary et al 2018 introduces the "F2" inversion approach explicitly to improve upon the crude assumptions of the "F1" inversion (page 5432: "The F1 function can help guide the search for the region where the presence of active source is most likely to concentrate, but of course the use of F1 alone does not represent a realistic distribution of sources in the MALM inversion.")

I think the authors should thoroughly explain why the "F1" inversion is sufficient in this study, or even better, they should provide "F2" results. This would bring this text more in line with the other paper. Otherwise this could be interpreted as the authors downgrading their previous approach.

—-

Line-specific comments:

4) line 39: what is the actual data that can be gained by the "new method"? In this regard, be more specific in lines 54/55: which data is required, which is extracted from MALM?

5) line 61: I think the "are" after "techniques" must be replaced by "is"

6) line 106: "test" -> "tested"

7) line 109: be more specific with regard to the field site and Mary et al 2018. I think it

strengthens the study if the link to the previous paper is made more specific.

8) line 114: as already stated, I'm missing this in the results/discussions

9) line 142: do you have any information on porosity/soil response to water content? Can you estimate an expected increase in resistivities due to the infiltration? Does the data fit any estimates?

10) line 165: can you be more specific on how you measure reciprocals for MALM? I would suspect totally different signal-to-noise environments, and correspondingly would deem the normal-reciprocal difference only as a weak proxy to data quality in this case, similar to nobody using normal-reciprocal differences for gradient or Schlumberger measurements.

11) section 2.4.1: can you provide error parameters used for the inversions? Is a target RMS of 1 reached for all time steps?

12) line 180: I suggest to rephrase and make clear that the voltage is measured with respect to the remote electrodes.

13) equations 1 (and others): could you indicate vector/matrix entities? For example, in equation 2 it is not clear if the F1-value is the norm of all misfits for a given current injection, or only for the i-th value (I suspect it is the former, but in this case the notation must be corrected).

14) line 186: I suggest to rephrase: the forward problem is unique, the inversion ill-posed. "relatively straightforward" is somewhat non-meaningful.

15) section 2.4.2 (MALM) lacks quite some details to properly understand the approach and it took me a while to figure out that the details can be found in Mary at al 2018 (not only the discussion of different approaches). While you link to Mary et al 2018 in line 189, perhaps you could start the section with a sentence similar to: "The MALM analysis follows Mary et al 2018..." to indicate that this is just a short recap and the reader cannot expect a comprehensive explanation here.

Again, I wonder why you chose to only use the F1 approach. You state in lines 201ff: "While more advanced attempts could be made (such as the F2 approach also described by Mary et al., 2018) the simple F1 approach is capable of imaging the likely location of current sources in the ground, that in turn represent - according to our key assumptions – the locations where roots have an active contact with the soil."

After reading through Mary et al 2018, I would expect a more detailed explanation, perhaps supported by numerical studies. Again, using only the F1 approach seems to contradict your findings in Mary et al 2018.

16) line 193: I'm not sure if the term "likelihood" is suitable here, given that it usually is associated with stochastic/Bayesian problem descriptions. Why not call it a data misfit, or a current source RMS?

17) section 3.1: Can you discuss more how you come to the conclusion that the intermediate depths are influenced by RWU (line 218-219)? Given the high irrigation rate of 115 l per hour (and ongoing infiltration), is the plant strong enough to take more water up than is infiltrated from above?

Are you sure that the high-conductive upper 40 cm in T1 are not cause by a somewhat higher porosity, compared to the layers below? (perhaps caused by the greater amount of roots reported in Mary et al 2018 for that layer?) Correspondingly, the anomalies in the "intermediate" region below could then be explained with a not-fully-saturated soil?

18) Section 3.2/Fig 4: I'm wondering if showing measured resistances helps here, due to varying geometric factors. What about showing the ratio of measured voltage and the homogeneous solution (Fig. 4e)? Alternatively, just convert to apparent resistivities?

19) line 250: "see Figure3Figure4" -> "see Figs. 3 and 4"?

20) line 251: perhaps prepare an appendix with the corresponding results for plant B?

21) line 260: I'm wondering how the selection of the F1-threshold using the 25% percentile influences the delineation of the "active" root zone. In Mary et al 2018 their

Figure 8 indicates possible threshold values of 35V (legend) or 17V (caption). Again, all for the same field site/plants. This seems inconsistent and should be discussed.

22) In relation to 21, it would be nice if you could try to actually recover root information that could, theoretically, be used in the SPAC-modeling approaches, as motivated in the introduction. Can the MALM-approach provide information on root density, or only on root delineation? Can you extract an exemplary 2D rooting depth/rooting density map from your results?

23) line 260: could you provide the curve of sorted F1-misfits in the appendix? This would ease the understanding of the F1-threshold.

24) line 268: I suggest to rephrase "absolute apparent"

25) line 277: the F1-threshold of 7V is used to delineate root and soil zones?

26) line 289: could you be more specific as to the resolution achieved here? My understanding is that the best resolution would be roughly the smallest electrode distance, which would be 10 cm here. Is this "high-resolution" in the context of root research?

27) line 294: "...independent information... may help". Could you be more specific on how these additional information could be included into your workflow, apart from validating the results? Would it be possible for you to include any of this data in your inversion workflow, thereby minimizing uncertainties?

28) lines 296-302: I cannot follow your reasoning regarding the "second coupling" of ERT and MALM. By design MALM requires ERT results, and thus I'm having problems seeing this as a conclusion.

29) line 303: I suggest to rephrase "successfully tested", as without validation we still do not know how reliable the results are.

30) lines 310-310: "The soil injection leads practically to identifying the true single electrode location." I'm having trouble understanding the meaning of the sentence...could

you elaborate or rephrase?

31) line 310: I don't get the meaning of the first sentence, given that ERT and MALM solve completely different inversion problems, which guarantees that the results differ. Perhaps rephrase with regard to the emerging patterns of the results?

32) Conclusions: I fail to find much discussion of the time-lapse character of the present study, and much overlap with the aims/conclusions of Mary et al 2018. Perhaps you could focus the conclusions more on time-lapse-specific questions/answers? Does MALM provide more robust information in the light of varying water content regimes? Can MALM provide information on the actual RWU, perhaps by correlating to the estimated evaporation?

33) Will you provide all primary data and analysis results as a data repository, as required by SOIL?

I sincerely hope these comments are not taken personally, but as constructive comments aimed a improving the message of the manuscript.

Best regards

References:

Mary, B., Peruzzo, L., Boaga, J., Schmutz, M., Wu, Y., Hubbard, S. S., and Cassiani, G.: Small-scale characterization of vine plant root water uptake via 3-D electrical resistivity tomography and mise-à-la-masse method, Hydrol. Earth Syst. Sci., 22, 5427-5444, https://doi.org/10.5194/hess-22-5427-2018, 2018.
* * *

---

## Referee Comment (RC2) · Anonymous Referee #2 · 11 Jun 2019

This manuscript deals with the application of geoelectrical methods for imaging and monitoring roots activity and water flow in the context of the soil-plant atmospheric continuum. The authors conducted time-lapse ERT and MLAM surveys around two grapevines plants differing in their age. The subsurface electrical resistivity was monitored before and after an infiltration experiment. The ERT data were inverted using a standard algorithm, and a simple algorithm for the imaging of the current source distribution was used. The work presented here is an extension of Mary et al (2018), but with the addition of an infiltration experiment, demonstrating the ability of the combined methods to monitor water content and RWU dynamics.

[Figure]

Overall, the work presented is interesting for the reader of SOIL, the manuscript is well written, and the methods and data analysis are adequate. The main pitfall is the lack of supplementary information that prevents a quantitative analysis of the (very interesting) dataset. Specifically, water content and water salinity were not measured or assessed. Differences between the transpiration of the two plants were not considered or measured (e.g., with sap flow meter). Nevertheless, even if this data is not available, the time laps MLAM provides qualitative information, at a high spatial resolution, on water content dynamics and RWU processes. In conclusion, I recommend publishing after some moderate revisions.

General comments:

1. The limitations in the interpretation of the results due to the lack of supplementary data (water content, salinity, formation factor) should be discussed in details.

2. In my view, one of the most interesting parts is the maps in Figure 5, showing the time-lapses differences between the young and old plants. However, a discussion on this observation is missing. Figure 5 is not mentioned in the text at all (perhaps in L252). I would strongly suggest to give a detail explanation of those results and to link them to the expected behavior of the different plants.

Specific comments: 1. L123: I guess that the water holding capacity is related to the pore size distribution and not to the porosity.

2. L142: you report the EC of the irrigation water, but what is the EC of the pore water? Do you expect heterogeneity in the pore water salinity? This should be discussed.

3. L223: Due to the lack of supplementary information, the arguments about the size and extent of the root systems are not solid enough. Is it the size of the root system or the total transpiration that differ?

Mary, B., Peruzzo, L., Boaga, J., Schmutz, M., Wu, Y., Hubbard, S. S., and Cassiani, G.: Small-scale characterization of vine plant root water uptake via 3-D electrical resistivity

tomography and mise-à-la-masse method, Hydrol. Earth Syst. Sci., 22, 5427-5444, https://doi.org/10.5194/hess-22-5427-2018, 2018.

---

## Referee Comment (RC3) · Anonymous Referee #3 · 14 Jun 2019

Review comments on Mary et al., 2019 SOIL This paper by Mary et al., proposed a novel and integrated geophysical monitoring framework to investigate the complex soil-root system, especially focusing on assessing the root water uptake and delineating the active root density. Such multidisciplinary and innovative research should be encouraged and supported as the authors are developing tools to provide quantifiable and potentially spatiotemporal intensive data for SPAC modeling. However, there are few major flaws in this paper that prevent it from publishing in its current form. I suggest the authors redesign the experiment, revise and expand the current manuscript according to the reviewers' comments, and resubmit it. Some general comments: 1. I assume this paper is meant to be an extension of Mary et al (2018) and to focus on infiltration

experiment. However, the datasets presented in this study and the affiliated discussions are not sufficient for a regular full paper, particularly, the lack of linking to any ground-truth data (such as soil samples, soil water chemistry, TDR measurements, rhizotron measurements, and so on). The authors also did not the full advantage of their >24 hours time-lapse measurements, only limited snapshots are presented without quantitative analysis. As a result, it is not convincing that this work has advanced the work from Mary et al., (2018), yet exhibits problematic overlaps. 2. In both current study and Mary et al., (2018), the biggest technical issue is that the electrode spacing is too small (0.1 m) and this might have violated the point-source assumption. The authors didn't explain what the electrodes they were using, or how deep the electrodes were buried in the ground. But from Figure 1 in this paper, it seems like authors used standard stainless steel electrodes with at least 10 cm into the ground (equal or even greater than the electrode spacing) This is extremely important as the current course in such setup (electrodes too close to the target and experiment dimension is on the same order as of the target) is very likely not 'point-source' anymore, and the noise could overwhelm the actual data due to target property changes. Such electrode mislocation errors can be very complicated but can be simulated in synthetic experiments. Furthermore, due to the principle of reciprocity, such data error cannot be caught and eliminated by reciprocal measurements. There are few studies on this problem and I strongly suggest the authors read related literature. I personally had failed experiments before due to this very reason. 3. The results and discussion are too brief and qualitative to provide an in-depth discussion on how the ERT and MALM reveal the actual root functions. For the readership of this journal, the actual root-soil mechanisms that were revealed and supported by geophysical methods are very appealing. The authors did a time-lapse (>24 hours) experiment, why the time-lapse ERT resistivity changes or MALM results are not shown? Only the initial condition and 2-hr snapshot are shown? More time-step data would provide significantly more information into the root system function. 4. More detailed soil information and geophysical survey design information should be provided. 5. An illustration showing the borehole locations is very necessary.

Also, please label the borehole number in the geophysical results plots as well. 6. Figure 5 shows the normalized voltage ratios for plant B, but this figure was not discussed or mentioned in the manuscript. 7. Figure 7 and the corresponding text section 3.4 are difficult to follow. First, where is the boundary of this estimated active root zone? What are the exact times from T1 – T5? Are these boxes representing all the ER values outside and inside the zone? Or just selected values? 8. Line 150. It is not very clear what is the electrode spacing for the surface electrodes, 0.1 m? what is the exact measurement configuration? The current description is too brief to get the idea of how the measurements were done (for example, any surface to borehole electrode pairs for current injection?) I've tried to read the Mary et al. 2018 paper, despite the similarity between these two studies, the ERT/MALM acquisition was not fully explained in that paper either. 9. Figure 3 needs to be improved with better visualization showing the 3D feature. The facets are not distinct in this current plotting style and the authors may organize the subplots into two rows for easier comparison.

Detailed comments: Line 35. Is the word 'expended' supposed to be 'expanded'? Line 36. SPAC is repeated. Line 37. I suggest more references here besides the work by Dirmeyer et al., Line 39. More references should be included. Line 55. Can the authors reiterate the main motivation of the work? Line 85-94. This part introduces the potential of SP and IP in monitoring water update and root systems. However, this part seems to be a bit out of place as the prior and following paragraphs discuss the actual methods have been used in this study. Suggest moving this part to either prior to ERT or after MALM. Line 209. 'less intense', what does this mean? Line 213 – 214. 'The input of low resistivity water (15 âĎęm, measured in laboratory) caused a homogeneous drop of 214 the resistivity values that make the two images around plant A and plant B very similar to each other'. How much is the resistivity decrease? Could you give a specific number? Maybe the authors can plot the delta resistivity (difference) for both plant A and B and show more time-step results. Figure 6. Please label 'stem/soil injection' directly on the plot to aid the reading.

---

## Editor Comment (EC1) · Paul Hallett (Editor) · 14 Jun 2019

Three reviewers have now provided very helpful and thorough comments on your manuscript. Reviewers 1 & 3 have concerns about the novelty of the study over Mary et al. (2018), which will need to be addressed with a strong and convincing statement about the originality of the submitted paper.

SOIL allows for public comment on all manuscripts so the paper will stay open until 3 July 2019, but you are welcome to comment on the reviews thus far.

---

## Author Comment (AC1) · 14 Oct 2019

**Paul Hallett (Editor)** paul.hallett@abdn.ac.uk

Three reviewers have now provided very helpful and thorough comments on your manuscript. Reviewers 1 & 3 have concerns about the novelty of the study over Mary et al. (2018), which will need to be addressed with a strong and convincing statement about the originality of the submitted paper.

SOIL allows for public comment on all manuscripts so the paper will stay open until 3 July 2019, but you are welcome to comment on the reviews thus far.

*Dear editor,*

*Thanks for giving us the opportunity to revise our article, and thanks for the reviews constructive comments. We took notice of all comments and the revised version of the manuscript.*

*In particular, we have reworked the manuscript in order to make its novelty more evident.*

*In order to deal with the lack of validation data, we have taken two actions:*
1. *We added a section describing a 1D hydrological modelling of infiltration;*
2. *We discussed the limitations of data to be used for validation, and added a section of recommendations for future experiments.*

*We agree with the reviewers' comments that the potential of this study and what makes its originality has not been fully exploited and explained. We took three actions in the revised version (after reformulating the goals):*
1. *We added the figures needed for a complete description and discussion of the full monitoring duration (30h) instead of the two related to the first-time steps;*
2. *We added an improved version of the algorithm of current density inversion applied to root location identification;*
3. *We discussed in detail the outcomes of the hydrological simulation that reproduce the infiltration experiment*

*A revised version of the manuscript is now being resubmitted.*
*Best regards,*
*Benjamin Mary et al.*

---

## Author Comment (AC2) · 15 Oct 2019

We thank the Reviewer for his/her comments. In the ensuing text, we try and address all raised issues. The reviewer's comments are reported in black, our replies in *italic blue*. Please also find attached a version of the manuscript with all changes highlighted in red.

B. Mary et al.
* * *
The manuscript "Time-lapse monitoring of root water uptake using electrical resistivity tomography and Mise-à-la-Masse: a vineyard infiltration experiment" by authors Mary et al investigates root characterization using the electrical MALM approach in a time lapse setting.

As explained in the manuscript, new approaches to non-, or minimally invasive root characterization are urgently needed to increase our knowledge about the root zone, as well as to provide better data input for soil-plant-atmosphere (SPAC) modeling frameworks. As such I think the topic of the manuscript is relevant to the readership of SOIL and well worth investigating. However, in its current state I cannot support a publication of the text under review without major revisions. Some parts of the manuscript feel somewhat rushed, and perhaps some (or most?) of my issues can be solved by reformulations or some additional text?

————— General Comments: —————

My two major concerns are:

1)      No validation data: The study uses neither independent information on the rooting depth or distribution, nor are soil information such as soil water content measurements, Archie-Parameters, or soil temperature data used to support the statements made (or used in the analysis of the data). The fact of missing validation data is also mentioned in the abstract, although not further discussed in the text.

*Validation of results on the basis of independent data is particularly challenging in the case of root water activity, and even worse in the case of field (not laboratory data). In order to deal with the lack of validation data, we took two actions:*

1. *We added the results of a 1D hydrological infiltration model. The simulation results are compared against the spatio-temporal changes of SWC obtained from ERT after petrophysical transformation using Archie's law. A new section was added in the methodology part (nb 2.4).*
2. *We discussed the limits of validation data in our case. See discussion section.*

In light of the novel, very promising, technical approach of MALM I don't think this should prevent the publication of the text, yet it should be actively discussed and conclusions should be limited to statements that can be made without validation data. Perhaps the text could be reworked to provide/develop recommendations for future experiments that deal with the problem of obtaining suitable validation data? As such, the direct predecessor paper, Mary et al 2018, and this study could be positioned as discussing the technical details of the approach, preparing future studies which focus on the validation aspect.

*Thanks for the suggestion. We feel this is indeed the message that should be conveyed by the paper. We added an explicit objective in this direction (see L. 123) and added a relevant paragraph in the discussion. Furthermore, all along the manuscript we moderated each of our interpretations acknowledging our lack of direct validation.*

Overlap with Mary et al 2018: As far as I can see Mary et al 2018 and this study were conducted on the same field site and same plants within a few months time. As such they should be considered as companion studies. As far as I can see their stated objectives overlap massively (as do the conclusions):

*Thanks for the suggestion to link the two papers using the companion paper option. We will deal with it for the re-submission.*

2)    Aims of Mary et al 2018 (page 5429): "1. define a viable field protocol that uses jointly MALM and ERT to map active tree vine roots, 2. propose and analyze algorithms capable of identifying the location of active roots, and 3. test the algorithms above against real data from a French vineyard."; Aims of this study: "(a) define a non-invasive investigation protocol capable of "imaging" the root activity as well as the distribution of active roots, at least in terms of their continuum description mentioned above; (b) Integrate the geophysical results with mass fluxes measurements in/out of the soil-plant continuum system." My reading is that all aims of Mary et al 2018 are contained in aim a) of this study.

*We agree with the reviewer comment, aims were reformulated to better highlight the specificities of this paper (see L. 119-123)*
*"*

*This study had the following goals:*

- *define a non-invasive investigation protocol capable of "imaging" the root activity as well as the distribution of active roots, at least in terms of their continuum description mentioned above, under varying soil water content conditions;*
- *integrate the geophysical results with mass fluxes measurements in/out of the soil-plant continuum system using a simple 1D simulation reproducing the infiltration experiment.*
- *give recommendations for future experiments focusing on the method validation.*

*"*

Adding to this, I was not able to find any information on the stated "integration of geophysical results with mass fluxes" (aim b) in the text, leaving only the duplicate aim a).

*See previous answer.*

Also note that the time-lapse aspect is currently not discussed in detail (detailed below) in the text, the analysis of the time-lapse data does not take into account any dynamics such as daily evaporation information, and the conclusion mostly reflects the conclusions of Mary et al 2018, without significant conclusions regarding the application of MALM within a time-lapse context.

*This concern has been also raised by reviewer 3. In this respect we took the following actions:*

- *Plot the full time-lapse data for ERT, MALM and current density for both plants and discussed it in detail (while this presentation/discussion was very limited in the first version of the manuscript). For those situations with no significant variations, plots are put into the appendix.*
- *The time-lapse variations are now discussed in light of the hydrological model. This allowed to clearly identify the dynamics such as daily evaporation and/or RWU.*

————————————- Specific comments/technical corrections: ————————————-
3)    In addition, there are various (apparent?) inconsistencies between Mary et al 2018 and this text. However, this could be caused by the rather brief formulations in the text, not by actual errors. I suggest to rephrase.

*We answered point by point hereafter.*

3.1) Site description: Mary et al indicate sandy-clayey soil from 125-175cm, while this text puts this layer from 100-175m. I also wonder why the authors do not interpret the different information on root distribution given in Mary et al 2018. For example, Mary et al 2018 identifies the first soil layer as "with a first sandy horizon (0–40 cm depth), porous and soft." Looking at Figure 3 in this text, I wonder if the observed resistivity decrease in the upper 40 cm can be attributed to this porous layer, and correspondingly fast infiltration?

*Simplifications in the text led to possible approximations but without any consequences on the data validity. At a depth larger than 100 cm the influence on our results is negligible, given the maximum depth of investigation of the electrode apparatus. We intended to describe the deepest layer only to highlight possible long-term water lift thanks to capillarity movements which could explain the shallow rooting depth hypothesized.*

*We rephrased the relevant sentences to make them consistent with the first article and we summarized the site description graphically in a new figure (as a support of the hydrological model)*

*The observed resistivity decrease in the upper 40 cm can be attributed to the porous layer, and correspondingly fast infiltration. We added a sentence to point it in the revised version of the manuscript L. 279*

*Note that in Mary et al 2018, the three layers were distinguished in term of root density and not in term of soil nature. Note that Figure 3 has been changed to highlight 3d feature as request by reviewer 3.*

3.2) Mary et al 2018 state that active roots are located in the upper 0.3m (page 5436). This does not seem to be the case if the isosurfaces in Figure 6 (this text) are to be interpreted as root extension, although the same plants are measured. Note that this text also states, as one of the conclusions, that the MALM approach is relatively insensible to different water regimes, and thus I would expect similar results between both studies.

If this inconsistency is caused by slightly different interpretation of the term "active roots", I suggest to rephrase accordingly.

*Obviously, the position of the active roots from one acquisition to the another (during different seasons) may vary. Sentence The sentence about 'active roots' has been rephrased to avoid misunderstanding.*

*First, in Mary et al. 2018, the value of 0.3 m has been validated using the target function F2, while the F1 plot is a spatial representation of the misfit (between measured data and individual source contribution). The misfit is normalized, but the threshold defined as the 25% percentile influencing the delineation of the "active" root zone is not directly comparable from one experiment to the other (see also answer for reviewer question nb 21).*

*Even though we think that function F1 might also be a good indicator for active roots, isosurfaces in Figure 6 of the manuscript mostly intend to show a straightforward comparison between soil and stem injection.*

3.3) Mary et al 2018 introduces the "F2" inversion approach explicitly to improve upon the crude assumptions of the "F1" inversion (page 5432: "The F1 function can help guide the search for the region where the presence of active source is most likely to concentrate, but of course the use of F1 alone does not represent a realistic distribution of sources in the MALM inversion.")

I think the authors should thoroughly explain why the "F1" inversion is sufficient in this study, or even better, they should provide "F2" results. This would bring this text more in line with the other paper. Otherwise this could be interpreted as the authors downgrading their previous approach.

*The reviewer is right, the F1 inversion is not sufficient per se to deduce root distribution but clearly convey the most interesting result of this article that is to say that the MALM results were corrected from variations of water content using ERT and thus are not sensitive to soil water content state which is the main problem for root detection using only ERT.*

*We took this comment into consideration and provide ALSO results from F2 inversion, and we took the opportunity to introduce some improvements as compared to the version used in the previous manuscript.*

*In the new version of the manuscript we adapted a robust algorithm to invert the data in 3D using the linearized form of the problem after Peruzzo et al. 2019 (Thesis.). The new features are (explained L. 246 to 260 of the revised version of the manuscript):*

- *Regularisation using a L-curve analysis to control the regularisation weight.*
- *The code has been written for the 3D case and for an unstructured mesh.*

*Most importantly all candidate sources are kept during the inversion of current density. Thus, there is no more a need to identify a threshold for which some sources are rejected. The misfit of F1 is transformed into a normalized initial model (m0) of current density via the inverse (1/F1) transformation. During the inversion of the current density, we adopted a relative smallness regularisation as a prior criterion for the inversion i.e. the algorithm minimizes $||m - m0||^2$, where m is the model parameter and m0 is a reference model to which we believe the physical property distribution should be close.*

*Result of F2 inversion are shown in the new version of the current manuscript.*

—-

Line-specific comments:

4) line 39: what is the actual data that can be gained by the "new method"? In this regard, be more specific in lines 54/55: which data is required, which is extracted from MALM?

*Sentence rephrased*

*In this study, we focus on new methods designed to image root systems and their macroscopic functioning, in order to help understand the complex mechanisms of these systems (the rhizosphere, e.g. York et al., 2016).*

5) line 61: I think the "are" after "techniques" must be replaced by "is"

*Ok done*

6) line 106: "test" -> "tested"

*Ok done*

7) line 109: be more specific with regard to the field site and Mary et al 2018. I think it strengthens the study if the link to the previous paper is made more specific.

*Ok done.  Sentence added: "This paper is meant to be an extension of Mary et al. (2018) and to focus on an infiltration experiment"*

8) line 114: as already stated, I'm missing this in the results/discussions

*See response to comment 3.1 and 3.2 above.*

9) line 142: do you have any information on porosity/soil response to water content? Can you estimate an expected increase in resistivities due to the infiltration? Does the data fit any estimates?

*These questions can now be answered thanks to the 1D hydrological modelling of the infiltration.*

*We here assumed that the retention and conductivity functions that describe the hydraulic properties of the soil can be represented by the Mualem-van Genuchten model (MVG, Mualem, 1976; van Genuchten, 1980.) Hydraulic properties of the soil were recovered directly from its grain size distribution – using Rosetta: see screen shot below.*

*Whilst this is a very simplified model, we were able to compare qualitatively the variations of resistivity with time with the infiltration simulation. To facilitate the comparison, electrical resistivities were transformed using Archie's law with the following parameters:*

- *porosity, assumed to be equal to soil $\vartheta s$ saturated water content;*
- *pore water electrical conductivity was assumed equal to the electrical conductivity of the water used for infiltration. The irrigation water had an electrical conductivity of 720µS/cm at 15°C. (0.072 S.m-1).*
- *m =1.3 (typical values notably described in Werban et al., 2008)*

[Figure]

10) line 165: can you be more specific on how you measure reciprocals for MALM? I would suspect totally different signal-to-noise environments, and correspondingly would deem the normal-reciprocal difference only as a weak proxy to data quality in this case, similar to nobody using normal-reciprocal differences for gradient or Schlumberger measurements.

*Study of reciprocals were more thoroughly discussed in the first paper Mary et al. 2018. It has been shown that reciprocals were indeed not easy to correlate with data quality in MALM. This may be caused by non-linearities caused during current injection in the stem itself.*

*We rephrased the sentence to better state that reciprocals may not be the best solutions to estimate data quality in the MALM case.*

11) section 2.4.1: can you provide error parameters used for the inversions? Is a target RMS of 1 reached for all time steps?

*We added a sentence to explain this in the text (L. 212 and 215 to 218) and completed the table 1.*

*Table 1 shows the performance of the inverse model in absolute mode such as number of rejected measurements, and final root mean square (RMS) for an error level of 10 %. Most of the data converged after fewer than 5 iterations. Error level was assumed to be equal to the error used for the reciprocal analysis.*

12) line 180: I suggest to rephrase and make clear that the voltage is measured with respect to the remote electrodes.

*Sentence rephrased, see L225*

13) equations 1 (and others): could you indicate vector/matrix entities? For example, in equation 2 it is not clear if the F1-value is the norm of all misfits for a given current injection, or only for the i-th value (I suspect it is the former, but in this case the notation must be corrected).

*Equation 1:*

*Poisson equation was conserved in its general form without description of the vector/matrix entities.*

*Equation 2 and 3 were further described See L. 241 and 252 to 260*

*The reviewer is right, F1-value correspond only to the i-th value.*

$F_1(D_m, D_{f,i}) = \|D_m - D_{f,i}\|_2$ *(2) changed to* $F_{1,i}(D_m, D_{f,i}) = \|D_m - D_{f,i}\|_2$

*Equation 3:*

$F_2 = \Phi_d + \lambda\Phi_m = W_\varepsilon\|d - f(m)\|_2^2 + \lambda(W_s\|m - m_0\|_2^2)$

14) line 186: I suggest to rephrase: the forward problem is unique, the inversion illposed. "relatively straightforward" is somewhat non-meaningful.

*Sentence rephrased as (see L. 230 to 233):*

*"*

*Given a distribution of current sources, and once $\sigma(x,y,z)$ is known from ERT inversion, the forward problem is uniquely defined and consists in the calculation of the resulting V field. Conversely, the identification of C(x,y,z) distribution given V(x,y,z) and $\sigma(x,y,z)$ is an ill-posed problem, that requires regularization and/or a priori assumptions in order to deliver stable results.*

*"*

section 2.4.2 (MALM) lacks quite some details to properly understand the approach and it took me a while to figure out that the details can be found in Mary at al 2018 (not only the discussion of different approaches). While you link to Mary et al 2018 in line 189, perhaps you could start the section with a sentence similar to: "The MALM analysis follows Mary et al 2018..." to indicate that this is just a short recap and the reader cannot expect a comprehensive explanation here.

*The sentence has been changed according to suggestion (see L. 222)*

Again, I wonder why you chose to only use the F1 approach. You state in lines 201ff: "While more advanced attempts could be made (such as the F2 approach also described by Mary et al., 2018) the simple F1 approach is capable of imaging the likely location of current sources in the ground, that in turn represent - according to our key assumptions – the locations where roots have an active contact with the soil." After reading through Mary et al 2018, I would expect a more detailed explanation, perhaps supported by numerical studies. Again, using only the F1 approach seems to contradict your findings in Mary et al 2018.

*The reviewer's comment makes sense. We took two actions:*

- *Plot the result of F2 inversion*
- *Rephrase the text to make it clear that F1 provides a feasible area of search to help F2 to converge (as explained in response to comments 3.2 and 3.3)*

15) line 193: I'm not sure if the term "likelihood" is suitable here, given that it usually is associated with stochastic/Bayesian problem descriptions. Why not call it a data misfit, or a current source RMS?

*Ok changed accordingly: Likelihood has been replaced by misfit*

16) section 3.1: Can you discuss more how you come to the conclusion that the intermediate depths are influenced by RWU (line 218-219)? Given the high irrigation rate of 115 l per hour (and ongoing infiltration), is the plant strong enough to take more water up than is infiltrated from above?

*The question "is the plant strong enough to take more water up than is infiltrated from above?" finds a convincing answer (YES) in a previous work conducted also on Citrus trees (Cassiani et al.,2015) where this phenomenon is apparent.*

*Note that figure 3 has been changed to show ER absolute values and ER ratios for the full-time monitoring.*

17) Are you sure that the high-conductive upper 40 cm in T1 are not cause by a somewhat higher porosity, compared to the layers below? (perhaps caused by the greater amount of roots reported in Mary et al 2018 for that layer?) Correspondingly, the anomalies in the "intermediate" region below could then be explained with a not-fully-saturated soil?

*This is a speculation that is not supported by the data we have – although the presence of dense roots may increase soil porosity (at a larger scale than a typical soil sample, though).*

18) Section 3.2/Fig 4: I'm wondering if showing measured resistances helps here, due to varying geometric factors. What about showing the ratio of measured voltage and the homogeneous solution (Fig. 4e)? Alternatively, just convert to apparent resistivities?

*For a pole-pole acquisition and even more for a mise-a-la-masse acquisition, results as usually displayed using horizontal maps of normalized **resistances – these are equivalent to voltages normalised on the injected current**. It has several advantages, in the perfect case, iso-potentials show directly the shape of the conductive body.*

19) line 250: "see Figure3Figure4" -> "see Figs. 3 and 4"?

*Corrected.*

20) line 251: perhaps prepare an appendix with the corresponding results for plant B?

*We have added an appendix for plant B*

21) line 260: I'm wondering how the selection of the F1-threshold using the 25% percentile influences the delineation of the "active" root zone. In Mary et al 2018 their Figure 8 indicates possible threshold values of 35V (legend) or 17V (caption). Again, all for the same field site/plants. This seems inconsistent and should be discussed.

*A value of the F1-threshold corresponding to the 25% percentile (here corresponding to 25% misfit) clearly influences the delineation of the "active" root zone as it can be appreciated from the misfit distribution below (time step T0 for the stem injection on the left, for the soil injection on the right).*

*See comment 23 for more details on the threshold determination.*

[Figure]

22) In relation to 21, it would be nice if you could try to actually recover root information that could, theoretically, be used in the SPAC-modeling approaches, as motivated in the introduction. Can the MALM-approach provide information on root density, or only on root delineation? Can you extract an exemplary 2D rooting depth/rooting density map from your results?

*SPAC modelling approaches integrating root water uptake use empirical model describing the root system (Dupuy et al. 2010) and MALM could indeed provide relevant information in this respect. Our approach is adaptable to macroscopic RWU models that describe root water uptake as a sink, microscopic ones being too complex since they required knowledges in root hydraulic architecture. This has been more extensively considered and discussed in this revised version of the manuscript.*

*Using the average value of current density along horizontal planes we were able to plot the vertical profile which was tentatively used as rooting profile density for the 1D hydrological simulation.*

*We discussed the limitations of MALM and the uncertainties on delineating the root system considering that only the active root system may be highlighted. In some cases, the active root system is really local as compared to the entire root system extent and thus the picture we can retrieve from excavation. As far as enough data are available to describe time varying*

*active root location MALM can improve macroscopic RWU modelling. Recommendations to use destructive methods to complement MALM methods were also introduced.*

23) line 260: could you provide the curve of sorted F1-misfits in the appendix? This would ease the understanding of the F1-threshold.

*We rephrased the sentence to make it clearer, but we did not overload the appendix with further figures. Below the curve of sorted misfit for T0 of plant B. See L. 320*

[Figure]

24) line 268: I suggest to rephrase "absolute apparent"

*Done*

25) line 277: the F1-threshold of 7V is used to delineate root and soil zones?

*Yes. We added a sentence to make it explicit. L. 345 to 348.*

26) line 289: could you be more specific as to the resolution achieved here? My understanding is that the best resolution would be roughly the smallest electrode distance, which would be 10 cm here. Is this "high-resolution" in the context of root research?

*Sentence rephrased*

27) line 294: "...independent information... may help". Could you be more specific on how this additional information could be included into your workflow, apart from validating the results? Would it be possible for you to include any of this data in your inversion workflow, thereby minimizing uncertainties?

*The first ERT-MALM coupling is already integrated in the workflow in the sense that we considered MALM as a non-soil water sensitive information to define the possible area of root. Separation of contributions of root zone and outer area on ER values extracted from ERT help distinguish between processes such as RWU and hydraulic redistribution (hydraulic lift in particular). This approach is more realistic compared to other approaches who used horizontally layered separation (Vanella et al.) to describe and separate the contributions.*

*The paragraph has been modified. See L. 384-393 of the new revised manuscript.*

28) lines 296-302: I cannot follow your reasoning regarding the "second coupling" of ERT and MALM. By design MALM requires ERT results, and thus I'm having problems seeing this as a conclusion.

*Sentence removed*

29) line 303: I suggest to rephrase "successfully tested", as without validation we still do not know how reliable the results are.

*Ok done*

30) lines 310-310: "The soil injection leads practically to identifying the true single electrode location." I'm having trouble understanding the meaning of the sentence...could you elaborate or rephrase?

*Sentence rephrased:*

*"*

*The soil injection leads practically to retrieve a current density close to a punctual injection (located at the true single electrode location)*

*"*

31)  line 310: I don't get the meaning of the first sentence, given that ERT and MALM solve completely different inversion problems, which guarantees that the results differ. Perhaps rephrase with regard to the emerging patterns of the results?

*Here we described the situation where the stem electrode is replaced by one in the soil but close to the stem (we are not referring to ERT!).*

*The sentence was rephrased to make it clearer.*

32)  Conclusions: I fail to find much discussion of the time-lapse character of the present study, and much overlap with the aims/conclusions of Mary et al 2018. Perhaps you could focus the conclusions more on time-lapse-specific questions/answers? Does MALM provide more robust information in the light of varying water content regimes? Can MALM provide information on the actual RWU, perhaps by correlating to the estimated evaporation?

*In the new version of the manuscript we try and correlate ER variations with estimated evapotranspiration and RWU. This also contributed to the second paper objectives which wasn't really fulfilled in the first version.*

*Questions "Does MALM provide more robust information in the light of varying water content regimes? Can MALM provide information on the actual RWU, perhaps by correlating to the estimated evaporation?" have been specifically addressed in the discussion section.*

33)  Will you provide all primary data and analysis results as a data repository, as required by SOIL?

*Yes. We added a statement on "Data availability" section.*

I sincerely hope these comments are not taken personally, but as constructive comments aimed a improving the message of the manuscript.

*We are grateful to the reviewer, for its thorough reading of the manuscript and for all the very relevant comments.*

Best regards

References:

**Cassiani G.**, J. Boaga, D. Vanella, M. T. Perri, S. Consoli, 2015, Monitoring and modelling of soil-plant interactions: the joint use of ERT, sap flow and Eddy Covariance data to characterize the volume of an orange tree root zone, *Hydrol. Earth Syst. Sci.*, 19, 2213-2225, doi:10.5194/hess-19-2213-2015.

Mary, B., Peruzzo, L., Boaga, J., Schmutz, M., Wu, Y., Hubbard, S. S., and Cassiani, G.: Small-scale characterization of vine plant root water uptake via 3-D electrical resistivity tomography and mise-à-la-masse method, Hydrol. Earth Syst. Sci., 22, 5427-5444, https://doi.org/10.5194/hess-22-5427-2018, 2018.

---

## Author Comment (AC3) · 15 Oct 2019

We thank the Reviewer for his/her comments. In the ensuing text, we try and address all raised issues. The reviewer's comments are reported in black, our replies in *italic blue*. Please also find attached a version of the manuscript with all changes highlighted in red.

B. Mary et al.

This manuscript deals with the application of geoelectrical methods for imaging and monitoring roots activity and water flow in the context of the soil-plant atmospheric continuum. The authors conducted time-lapse ERT and MLAM surveys around two grapevines plants differing in their age. The subsurface electrical resistivity was monitored before and after an infiltration experiment. The ERT data were inverted using a standard algorithm, and a simple algorithm for the imaging of the current source distribution was used. The work presented here is an extension of Mary et al (2018), but with the addition of an infiltration experiment, demonstrating the ability of the combined methods to monitor water content and RWU dynamics. Overall, the work presented is interesting for the reader of SOIL, the manuscript is well written, and the methods and data analysis are adequate.

The main pitfall is the lack of supplementary information that prevents a quantitative analysis of the (very interesting) dataset. Specifically, water content and water salinity were not measured or assessed. Differences between the transpiration of the two plants were not considered or measured (e.g., with sap flow meter). Nevertheless, even if this data is not available, the time lapse MALM provides qualitative information, at a high spatial resolution, on water content dynamics and RWU processes. In conclusion, I recommend publishing after some moderate revisions.

General comments:

1. The limitations in the interpretation of the results due to the lack of supplementary data (water content, salinity, formation factor) should be discussed in details.

*We acknowledge the limited availability of supporting data. However, in order to strengthen the paper's conclusions we took two actions:*

  a) *We added a 1D hydrological modelling of the infiltration:*

*Whilst this is a simplified model, we used a petrophysical transformations (Archie) on measured ER and thus recovered the spatial and time variations of soil water content.*

  b) *We discussed the limits of validation data in our case:*

*We do agree that ancillary measurements are always welcome to support the geophysical information. Nevertheless, in this specific case they also have important limitations:*

  - *Validation through root excavation has numerous potential pitfalls. Among them, the destruction of fine roots during extraction that may prevent us from correlating root system architecture et geophysical observations. Discussed L417 to 423*
  - *SWC can only be measured at few specific spatial locations. Discussed L. 425*

2.      In my view, one of the most interesting parts is the maps in Figure 5, showing the time-lapses differences between the young and old plants. However, a discussion on this observation is missing. Figure 5 is not mentioned in the text at all (perhaps in L252). I would strongly suggest to give a detail explanation of those results and to link them to the expected behavior of the different plants.

*The reviewer might have misunderstood the meaning of the Fig. 5. Maps in Figure 5 showed differences between the MALM stem injection and its companion soil injection (as explained in the section MALM acquisition 2.3). Also, reviewer 3 reported that the figure was redundant with Fig. 4 and was not sufficiently mentioned. We then decided to remove it and improve Fig. 4 instead to better convey the idea i.e. raw map of MALM potential contains time lapse information and show significant differences between stem and soil injection.*

*As for the comparison between plants A and B, this has not been neglected since in the initial version of the manuscript we discussed the differences between the two plants (see L. 265/266, 275/276, 314, 319, …).*

Specific comments:

1.      L123: I guess that the water holding capacity is related to the pore size distribution and not to the porosity.

*Correct. We rephrased the sentence accordingly.*

*New sentence: "*

*Due to its larger particles and thus high porosity and smaller surface area, the sandy layer has a relatively poor water retention capacity.*

*" See L. 132 of the revised manuscript*

2.      L142: you report the EC of the irrigation water, but what is the EC of the pore water? Do you expect heterogeneity in the pore water salinity? This should be discussed.

*For the hydrological model reproducing the infiltration test, pore water conductivity was assumed equal to electrical conductivity of the water used for infiltration.*

*We have good reason to think that this assumption can hold in our case since the infiltration was relatively intensive (> 100L/h) and the initial soil pores were then filled with irrigation water. We nevertheless agree with the reviewer comment that EC of the pore water is not necessarily equal to the EC of the irrigation. We did not consider specific rhizosphere processes such as root exudation, which could affect the water content estimates. We assumed, no salt accumulation taking place near the roots, i.e. passive solute uptake only with no active uptake, exclusion or exudation. Solute movement models exist to consider the different root processes that might affect the constant concentration in water pores. Significant solute gradients may arise around roots due to processes mentioned above. If they were to occur, the water content estimates could be impacted by such gradients. However, in this specific case the volume of irrigated water and the short residence time is likely to make these processes second order effects.*

3.      L223: Due to the lack of supplementary information, the arguments about the size and extent of the root systems are not solid enough. Is it the size of the root system or the total transpiration that differ?

*If the reviewer is talking about the differences between the shape of the two plants, we do agree that a pure comparison of their root system is not realistic from our result. Nevertheless, this is clearly not the point of this study.*

*Nevertheless, the reviewer raised an interesting question that we tried to address in the discussion (see section 4.1 of the revised manuscript)*

---

## Author Comment (AC4) · 15 Oct 2019

We thank the Reviewer for his/her comments. In the ensuing text, we try and address all raised issues. The reviewer's comments are reported in black, our replies in *italic blue*. Please also find attached a version of the manuscript with all changes highlighted in red.

B. Mary et al.
* * *
Review comments on Mary et al., 2019 SOIL This paper by Mary et al., proposed a novel and integrated geophysical monitoring framework to investigate the complex soil root system, especially focusing on assessing the root water uptake and delineating the active root density. Such multidisciplinary and innovative research should be encouraged and supported as the authors are developing tools to provide quantifiable and potentially spatiotemporal intensive data for SPAC modeling. However, there are few major flaws in this paper that prevent it from publishing in its current form. I suggest the authors redesign the experiment, revise and expand the current manuscript according to the reviewers' comments, and resubmit it. Some general comments:

1. I assume this paper is meant to be an extension of Mary et al (2018) and to focus on infiltration experiment. However, the datasets presented in this study and the affiliated discussions are not sufficient for a regular full paper, particularly, the lack of linking to any ground-truth data (such as soil samples, soil water chemistry, TDR measurements, rhizotron measurements, and so on). The authors also did not the full advantage of their >24 hours time-lapse measurements, only limited snapshots are presented without quantitative analysis. As a result, it is not convincing that this work has advanced the work from Mary et al., (2018), yet exhibits problematic overlaps.

*Indeed, this can be viewed as a companion paper of the previous paper by Mary et al. (2018) – the same was suggested by Rev.1; note that the results contained in both papers are different, complementary to each other, and too extensive to be summarised in a single paper, referring also to different experiments. Given this, we reformulated the objectives of this paper to better convey its originality and importance as compared to the paper by Mary et al. (2018):*

*The objectives are:*
- *define a non-invasive investigation protocol capable of "imaging" the root activity as well as the distribution of active roots, at least in terms of their continuum description mentioned above;*
- *integrate the geophysical results with mass fluxes measurements in/out of the soil-plant continuum system using a simple 1D simulation reproducing the infiltration experiment.*
- *give recommendations for future experiments that deal which focus on the validation aspect.*

*Furthermore, in the revised version of the manuscript we took care of showing the full advantage of the time-lapse measurements by:*
- *showing the time-lapse variation of absolute ER inverted values for all time steps*
- *inverting and showing the time-lapse ratios after time lapse inversion of ERT data*
- *showing the time-lapse variation of MALM for all time steps*

*Also, in the revised version and in order to comply with the new objectives, we present the results of a 1D simulation of the infiltration experiment. The time-lapse results are now discussed in the light of the hydrological model. This allowed to clearly identify the dynamics such as daily evaporation and/or RWU.*

*Finally, in the revised version of the manuscript (also considering the comments by Reviewer 1) we also improved the inversion algorithm in the F2 formulation.*

2. In both current study and Mary et al., (2018), the biggest technical issue is that the electrode spacing is too small (0.1 m) and this might have violated the point-source assumption. The authors didn't explain what the electrodes they were using, or how deep the electrodes were buried in the ground. But from Figure 1 in this paper, it seems like authors used standard stainless steel electrodes with at least 10 cm into the ground (equal or even greater than the electrode spacing) This is extremely important as the current course in such setup (electrodes too close to the target and experiment dimension is on the same order as of the target) is very likely not 'point-source' anymore, and the noise could overwhelm the actual data due to target property changes. Such electrode mislocation errors can be very complicated but can be simulated in synthetic experiments. Furthermore, due to the principle of reciprocity, such data error cannot be caught and eliminated by reciprocal measurements. There are few studies on this problem and I strongly suggest the authors read related literature. I personally had failed experiments before due to this very reason.

*The reviewer is right at raising the question but is wrong in his/her conclusions that we used standard electrodes buried as deep as 10 cm. We are perfectly aware of the issues related to the assumption of point sources inherent in most inversion algorithm, and we have 20 year experience in small scale ERT applications where these issues are constantly taken into account – the literature is well known. In this particular case the risk is invariably that of having an apparent conductive layer at the surface. And the impact in time-lapse measurements tends to be factored out, of course.*

*Nevertheless, not that the surface electrodes are 1.4 mm in diameter, but are stuck in the ground by no more than 3 cm – note that the soil surface is very irregular between the rows of the vineyard due to field work of the land surface.*

3. The results and discussion are too brief and qualitative to provide an in-depth discussion on how the ERT and MALM reveal the actual root functions. For the readership of this journal, the actual root-soil mechanisms that were revealed and supported by geophysical methods are very appealing. The authors did a time-lapse (>24 hours) experiment, why the time-lapse ERT resistivity changes or MALM results are not shown? Only the initial condition and 2-hr snapshot are shown? More time-step data would provide significantly more information into the root system function.

*Thanks for the suggestion. In the revised version of the manuscript, the time-lapse ERT resistivity changes, derived from a time-lapse ratio inversion and MALM results for all time steps and for both plants are presented (plant B shown only in appendix for brevity). Also, a dedicated section was added to discuss integration of ERT with the hydrological model and ET data.*

4. More detailed soil information and geophysical survey design information should be provided.

*Soil information (type, roots density and granulometry) is now reported in the new figure 2a and a detailed scheme of the survey design is now added in appendix A1.*

5. An illustration showing the borehole locations is very necessary. Also, please label the borehole number in the geophysical results plots as well.

*See previous comment. All the electrodes numbering is now labelled in the figures.*

6. Figure 5 shows the normalized voltage ratios for plant B, but this figure was not discussed or mentioned in the manuscript.

*Figure 5 has been removed from the new version of the manuscript because the raw MALM data does not provide a straightforward information. We added in the appendix the time-lapse variation of the absolute normalised voltage measured.*

7. Figure 7 and the corresponding text section 3.4 are difficult to follow. First, where is the boundary of this estimated active root zone? What are the exact times from T1 – T5? Are these boxes representing all the ER values outside and inside the zone? Or just selected values?

- *Our assumption is that the region identified by MALM F1 (albeit very rough) for the background time corresponds to the RWU region. The inner area (IN) is then defined as the area within the closed isosurface at the background time T0.*
- *The times for T1 to T5 are given in the table 1 (note that there is no exact time since the measurement last approximately 30min. Legend has been rephrased to stressed out this (We invite the reader to report to this table otherwise the figure would be overloaded).*
- *Boxes represent selected values inside and outside the hypothetic rooted zone. Left size figure boxes refer to the OUT zone while right size to the IN. Figure and figure legend were improved to better convey this.*

*For the reviewer information we show hereafter a plot the boundary of the active zone below to illustrate our comment.*

[Figure]

8. Line 150. It is not very clear what is the electrode spacing for the surface electrodes, 0.1 m? what is the exact measurement configuration? The current description is too brief to get the idea of how the measurements were done (for example, any surface to borehole electrode pairs for current injection?) I've tried to read the Mary et al. 2018 paper, despite the similarity between these two studies, the ERT/MALM acquisition was not fully explained in that paper either.

*In the revised version of the manuscript we detailed the set-up geometry adding a figure in appendix A1.*

*We also added a more in-depth description of the protocol (see L. 165).*
*"*
*The total dataset includes three types of measurements: 430 surface-to-surface, 2654 surface-to-borehole and 4026 in-hole measurements.*
*"*

9. Figure 3 needs to be improved with better visualization showing the 3D feature. The facets are not distinct in this current plotting style and the authors may organize the subplots into two rows for easier comparison.

*The new figure shows now the 3D pattern via a combination of vertical and horizontal slices through the 3D interpolated ER, slice positions were chosen to correspond to the control point of the hydrological simulation.*

Detailed comments:

Line 35. Is the word 'expended' supposed to be 'expanded'? *Corrected*
Line 36. SPAC is repeated. *Deleted*
Line 37. I suggest more references here besides the work by Dirmeyer et al.,

*Added a reference to Newman et al., (2006).*
*"Newman, B. D., Wilcox, B. P., Archer, S. R., Breshears, D. D., Dahm, C. N., Duffy, C. J., McDowell, N. G., Phillips, F. M., Scanlon, B. R. and Vivoni, E. R.: Ecohydrology of water-limited environments: A scientific vision: OPINION, Water Resour. Res., 42(6), doi:10.1029/2005WR004141, 2006."*

Line 39. More references should be included.

*We added a sentence and a reference to Richter and Mobley, (2009).*
*"Richter, D. deB. and Mobley, M. L.: Monitoring Earth's Critical Zone, Science, 326(5956), 1067–1068, doi:10.1126/science.1179117, 2009."*

Line 55. Can the authors reiterate the main motivation of the work?

*Sentence rephrased:*
*"*

*However, calibration requires that suitable data such as roots and soil water content evolution are available in a form comparable with the model to be calibrated.*

*"*

Line 85-94. This part introduces the potential of SP and IP in monitoring water update and root systems. However, this part seems to be a bit out of place as the prior and following paragraphs discuss the actual methods have been used in this study. Suggest moving this part to either prior to ERT or after MALM.

*Thanks. Done*

Line 209. 'less intense', what does this mean?

*"Intense" replaced by "resistive"*

Line 213 – 214. 'The input of low resistivity water (15 Ohm.m, measured in laboratory) caused a homogeneous drop of the resistivity values that make the two images around plant A and plant B very similar to each other'. How much is the resistivity decrease? Could you give a specific number? Maybe the authors can plot the delta resistivity (difference) for both plant A and B and show more time-step results.

*Thanks for the suggestions.*
*In the revised version of the manuscript we added the time-lapse inversion to evaluate the resistivity decrease or increase in term of % of change on the ratios between two consecutive times for both plants.*
*Absolute change of ER is about 50 Ohm.m and up to 100 Ohm.m  (added in the revised manuscript L. 278)*

Figure 6. Please label 'stem/soil injection' directly on the plot to aid the reading.

*Done*

---

## Author Comment (AC5) · 15 Oct 2019

The comment was uploaded in the form of a supplement:
https://www.soil-discuss.net/soil-2019-28/soil-2019-28-AC5-supplement.pdf

---

## Author Comment (AC7) · 15 Oct 2019

The comment was uploaded in the form of a supplement:
https://www.soil-discuss.net/soil-2019-28/soil-2019-28-AC7-supplement.pdf

---

## Editor Decision (ED1)

**Manuscript soil-2019-28 – Minor revision**
**B. Mary et al.**
*Time-lapse monitoring of root water uptake using electrical resistivity tomography and Mise-à-la-Masse: a vineyard infiltration experiment*

Dear Editor, dear reviewers,

We thank again the reviewers for their careful reading.

In the revised version of the manuscript text, we addressed all raised minor changes. Please also find attached a version of the manuscript with all changes highlighted.

Data are now available on the open data repository of the University of Padova.

Based on these changes, we would be grateful if you would consider the revised manuscript for publication in Soil.

Best regards,

*B. Mary et al.*

Review of the revised version of: "Time-lapse monitoring of root water uptake using electrical resistivity tomography and Mise-à-la-Masse: a vineyard infiltration experiment" by Benjamin Mary et al.

The main revisions in the revised version of the manuscript are: (1) addition of a 1D hydrological model of the infiltration; (2) an extended discussion on the limited availability of supporting information; (3) reformulation of the objectives; (4) extended discussion on time-lapse data. Overall, I think that the authors did a good job addressing the reviewers' comments and improved the manuscript significantly. Attached are some comments, which after their consideration, I believe the manuscript can be accepted for publication.

1. L160 – please also give the electrode length

*Sentence rephrased "a better contact in the loose soil and were heavier and more firmly grounded (3cm out of 12)"*

2. L187 – It is not clear enough how you process the results from the MALM to obtain a 1D root length density. Please explain how that was achieved.

*The reader should refer to L. 197 for more details. Sentence rephrased.*

3. Table 1– Instead of the number of data points used for the inversion, can you please give the percentage of data that passed the reciprocity error threshold?

*Ok done*

4. L217-218 – Can you provide information about the % of data that did not pass the 2% error?

*Ok done. Sentence rephrased.*

*At this threshold 65% (in mean) of the data passed the reciprocity. A total number of 687 points were used during the inversion after selection of common set between all-time steps.*

5. L261 and Fig 3. – In the text you refer to the wrong Fig. please correct. In Fig. 3 you have wrong references to the a and b panels.

*Ok done*

6. Make sure to correct all the references to Fig. 3 (e.g., L261, L263, 264)

*Ok done*

7. L275 – correct the ref. to the figure.

*Ok done*

8. In Fig. 5, the graphs for the boreholes. It is not clear what each of the lines means, i.e., which fig. is related to a specific borehole.

*Ok we added a legend to identify the boreholes numbering*

9. Rephrase L340-341

*Done*

10. L 356 – missing Ref to Figure

*Refs added*

11. Section 3.5 – Plotting 1D profiles of the resistivity (say under the root zone) can help in the comparison between the observed and modeled dynamics.

*Since we modelled and show only the variations of soil water content we limited the plots to the 1d profiles of converted resistivity to SWC.*

12. L401 – missing "in" ( "results ____ a simple")

*Ok thanks*

Dear authors, dear editor, the manuscript "Time-lapse monitoring of root water uptake using electrical resistivity tomography and Mise-à-la-Masse: a vineyard infiltration experiment" describes electrical tomographic measurements on two vine plats during an infiltration experiment. Here, the MALM method is further investigated as a means to infer the distribution of active root distributions, based on the basic premise that electrical current, injected into the stem of plants, follows the root elements down to their endings, thereby forming a relationship between root (end) distribution and electrical measurement characteristics.

This is the first revision of the manuscript -- I also reviewed the initial submission.

In general, I think the authors substantially improved the manuscript and shaped the discussion of the available data. I also think that the MALM method, coupled to imaging, will see substantial activity in the upcoming years, which will make this manuscript part of a substantial base of knowledge from which to investigate further.

Yet, I think some (minor) aspect still could use some polishing, and in general I found a small number of inconsistencies/mistakes. Therefore, I suggest a minor revision to provide enough time for polishing the presentation and go over the text again.

Comments:

- line 96/97: in this regard the study of Rao et al, 2019 would be a nice addition to the references, dealing with exactly this type of influence.

https://doi.org/10.2136/vzj2019.04.0037

*Done*

- line 145/146: I'm not a soil scientist, but based on the soil description (sand), the generally low precipitation in the month preceding the experiment, and the high air temperature, I would expect the SWC to be below field capacity, and more approaching the wilting point (my reasoning: the reported 18 mm cumulative precipitation in the month before would be gone after 4 days at 5 mm/day, and even ignoring runoff and drainage, at this point SWC would fall below field capacity).

*We agree with the reviewer comment and replaced close by below*

- line 160: electrodeS -> electrode

*Done*

- line 161: I'm not sure I understand the notation of "(3/cm)"

*Corrected (3cm out of 10)*

- line 168: support -> supported

*Done*

- line 171: "remoteS" -> "remote"

*Done*

- line 187: result -> resultS

*Done*

- What are the Archie-Parameter used? What is the assumed porosity distribution? I understand that you provide some of the answers in the review-reply, but would strongly suggest to include them also in section 2.4.

*We added the missing Archie-Parameters used and the assume porosity distribution*

*The porosity was assumed to be equal to the soil saturated water content ($\vartheta s$), the cementation factor (m) equal to 1.3 and the saturation exponent (n) equal to 1 (typical values notably described in Werban et al., 2008).*

- eq (1), comma after equation

*Done*

- eq (2): I would suggest to differentiate vectors by printing them bold (as done in line 228 for C).

Similarly, it would be nice to also mark matrices differently (eq. 3). My suggestion would be to write vectors and matrices in bold, with lower characters assigned to vectors, and upper characters to matrices.

*Done for eq. 2 and 3*

- eq 3: I believe that $W_{eps}$ and $W_s$ need to be moved into the norms (this would be consistent with standard least-squares inversion theory). Also, currently you try to apply a Matrix ($W_{eps}/W_s$) to a single value (the norms). End equation 3 with a point (end of sentence)

*Done*

- is there any normalization included for the $F_2$ inversion? I.e., does the injected current always sum up to 1 Ampere (or any other normalization constant?).

*Yes, we applied the current conservation law to normalise F2. Sentence added "Lastly, current conservation was respected since the sum of cj was equal to 1 at the end of the inversion iterations."*

- section 2.5.2 in general: Reading just this section, I get the strong sense that the F2 inversion is the one that should actually be analyzed, and F1 is only used to determine a suitable reference/starting model (lines 254++). Yet, later on the F1-result are still prominently analysed (3.3/3.4), while the F2 inversion results are only shortly discussed lines 336-341. I suggest to adjust the formulations accordingly.

*We think that function F1 might also be a good indicator for active roots and that a straightforward comparison between soil and stem injection can be achieved with it. With our methodology, F1 provides a feasible area of search to help F2 to converge and the extension of F1 was very similar to F2. Yet this assumption is not supported by numerical studies but we would like to investigate it to offer a simple way to process MALM data without having to go through an inversion. This line of research has been initiated with the contributions of Binley et al (Late 90's)\**

- Rephrase the text to make it clear that line 264: citation error in pdf

*Corrected*

- Fig 3: increase size of colorbar

*Done*

- line 261: figure references should be figs 3 and 4, I believe

*Corrected*

- lines 267 - 272: These three sentences are inconsistent. The first one discuss a low resistive layer (consistent with ERT figures), but the second and third one actually argue for increased resistivities using RWU. :-)

*The explication for the increased resistivities using RWU related to resistive anomalies at intermediate depths. Sentence relocated for more consistencies.*

- lines 203 vs 275: the conductivity of irrigation water is inconsistent between both lines. Line 203 mentions a conductivity of 720 mu S/cm, which I believe convert to 13.88 Ohm m, while line 275 gives 15 Ohm m.

*Corrected*

- line 280: Figure reference misses the figure number (should be fig 4?)

*Yes done*

- Fig 4b: To be honest, I find the values confusing. Why not show percentage changes with respect to T0? But I suppose this is a matter of personal taste, so please ignore.
- eq 4 (and sentence after): comma after equation. Also, I would suggest to add a note that this is only valid for pole-pole measurements, to prevent any confusions.

*We rephrase the previous sentence to make clear that this is the voltage distribution due to a single current electrode*

- Graphics quality in general should be checked (could be a matter of the review pdf)

*All figures are now at least produced with 300dpi.*

- lines 286-294: I suggest to only talk in terms of resistances - it is can be quite confusing to first read eq 4, and then only see resistances in Fig 5 (this is a presentation thing, you correctly mention the normalization).

*Done for this paragraph*

- caption of Fig 5/line 625: here equation 2 is mentioned, yet you show raw data and I believe 5e depicts computations with eq. 4?

*True thanks. Corrected*

- Fig 5: relate colors of vertical plots to electrode boreholes (e.g., to electrode numbers in Fig. 7)

*Done*

- line 346: missing Figure reference number

*Done ref added*

- Fig. 8: what are the dots? Data points determined as outliers?

*Yes, added in the legend for clarity.*

- line 367: showS -> show

*Done*

- line 429: perhaps replace "next" by "near"?

*Done*

- section 4.2 reads very pessimistic, and implies that traditional methods are not reliable and cannot be used for validation of MALM. I would argue that traditional root sampling methods, although labour intensive and with known limitations regarding fine root detection, should be the first line of validation, given that soil/root sciences have been working with this data for decades. Only in the second line of argumentation should alternative means be discussed. Also, in the case of vine plant roots, which I believe are of the woody nature, even a destructive sampling by trench method should provide a pretty good idea of root distribution. Coupled with knowledge about water distribution by traditional ERT, the intersection of both distributions should provide us with a pretty good first estimate of where RWU can be expected (and such help to validate MALM results).

*Ok sentence moderated accordingly*

- - In general, I propose to rework the figures for:

a) consistent styling/layout b) proper figure sizes (I believe that final figures should be either 8.3 cm (single column) or 12 cm in width, and it seems to me that a lot of figures will have really small/unreadable text after adjusting for these sizes.

b) fix overlapping lables (e.g., fig 5).

*Done*

- line 694, caption of Fig C4: descriptions of left and right panel are swapped

*Corrected*

Looking forward to seeing the published version

*Thanks!*

Best regards

[revised manuscript text omitted]